# A new Paleogene fossil and a new dataset for waterfowl (Aves: Anseriformes) clarify phylogeny, ecological evolution, and avian evolution at the K-Pg Boundary

**Grace Musser** [1,2]*, **Julia A. Clarke**[2]

**1** Department of Vertebrate Zoology, Division of Birds, The Smithsonian National Museum of Natural History, Washington, District of Columbia, United States of America, **2** Department of Earth and Planetary Sciences, The University of Texas at Austin, Austin, Texas, United States of America

* musserg@si.edu

**Data Availability Statement:** CT slices have been deposited on DRYAD under the following DOI: 10. 5061/dryad.v15dv4208. All other Supplementary Data is available on Morphobank under Project

## Abstract

Despite making up one of the most ecologically diverse groups of living birds, comprising soaring, diving and giant flightless taxa, the evolutionary relationships and ecological evolution of Anseriformes (waterfowl) remain unresolved. Although Anseriformes have a comparatively rich, global Cretaceous and Paleogene fossil record, morphological datasets for this group that include extinct taxa report conflicting relationships for all known extinct taxa. Correct placement of extinct taxa is necessary to understand whether ancestral anseriform feeding ecology was more terrestrial or one of a set of diverse aquatic ecologies and to better understand avian evolution around the K-T boundary. Here, we present a new morphological dataset for Anseriformes that includes more extant and extinct taxa than any previous anseriform-focused dataset and describe a new anseriform species from the early Eocene Green River Formation of North America. The new taxon has a mediolaterally narrow bill which is rarely found in previously described anseriform fossils. The matrix created to assess the placement of this taxon comprises 41 taxa and 719 discrete morphological characters describing skeletal morphology, musculature, syringeal morphology, ecology, and behavior. We additionally combine the morphological dataset with published sequences using Bayesian methods and perform ancestral state reconstruction for select morphological, ecological and behavioral characters. We recover the new Eocene taxon as the sister taxon to (Anseranatidae+Anatidae) across all analyses, and find that the new taxon represents a novel ecology within known Anseriformes and the Green River taxa. Results provide insight into avian evolution during and following the K-Pg mass extinction and indicate that Anseriformes were likely ancestrally aquatic herbivores with rhamphothecal lamellae..

## Introduction

Although extant Anseriformes appear to occupy a relatively narrow range of dominantly aquatic ecologies, both extinct and extant taxa show a broad range of locomotor and feeding

4001 at the following link: http://morphobank.org/permalink/?P4001.

**Funding:** This project was supported by a National Science Foundation GRFP (https://www.nsfgrfp.org/) award (to G.M., grant number DGE-16-4486), an Ornithology Collections Study Grant from the American Museum of Natural History (https://www.amnh.org/research/vertebrate-zoology/ornithology ;to G.M., 2019) and the Jackson School of Geosciences (https://www.jsg.utexas.edu/ ;G.M. and J.A.C). The funders had no role in study design, data collection and analysis, decision to publish, or preparation of the manuscript.

**Competing interests:** The authors have declared that no competing interests exist.

modes. Extinct ecotypes exhibited in the comparatively rich Cretaceous-Paleogene fossil record of this clade include the terrestrial, giant flightless birds within *Gastornis* [1,2, *vide* 3–5] with dorsoventrally broad beaks and debated diets [6–8]; the marine albatross-like Pelagornithidae with pseudotooth projections for piscivorous diets [9–11]; and the more aquatic, duck-like species such as the wide-billed, but long-legged wader *Presbyornis* [12–14]. Extant Anseriformes include the largely terrestrial Anhimidae that are primarily herbivorous, semi-aquatic foragers; *Anseranas*, an aquatic surface swimmer and grazer that is primarily herbivorous; and a variety of ecotypes within Anatidae [15,16]. Within Anatidae, taxa possess both herbivorous and omnivorous diets with various degrees of specialization; different filter feeding modes comprising grazing, mixed feeders, and diving graspers; and a variety of swimming ecologies that comprise a terrestrial ecology, foot propelled swimming, wing propelled swimming, surface swimming, plunging, and foot and wing propelled swimming [17–19]. Diving, as either an escape behavior or in feeding, occurs throughout Anatidae but not in Anhimidae and possibly not in *Anseranas* [16,20,21].

Despite the ecological breadth of anseriform ecologies and an abundance of recovered extinct taxa, important questions remain regarding the phylogeny and the ecological and behavioral evolution of Anseriformes. It is debated when within stem or crown Anseriformes they evolve more aquatic ecologies, especially as vestigial rhamphothecal lamellae and semipedal webbing were proposed to be present in extant Anhimidae [22]. It is similarly uncertain how herbivory and beak shape evolved within this group. Simulated trait evolution has supported two main patterns of diversification of beak shape and related diet in crown Anseriformes: either a single evolutionary trajectory or several independent and parallel transitions to a narrower, more herbivorous "goose-like" beak are estimated [19]. Phylogenetic placement of extinct anseriform taxa is necessary to inform evolution of these traits in both the stem and the crown.

Morphological analyses containing extinct taxa have resulted in drastically differing topologies [11,13,14,23–28]. Additionally, specimens that represent or are referrable to several extinct taxa have not been included or fully captured by scorings in previous analyses, resulting in a higher level of missing data in these analyses. At the same time, fine-grained molecular analyses of extant taxa have produced conflicting results within Anatidae [29–31]. Robust placement of extinct taxa is critical as ancestral state reconstructions are optimized differently depending on variations in extinct taxon placement. Better understanding the evolutionary relationships and ecological evolution of Anseriformes and their stem lineages is also necessary for better understanding early avian evolution and biogeography. New Paleogene fossils and robust placement of these and previously described taxa within a phylogenetic context is needed to better understand the answers to these questions.

The predominantly lacustrine Green River Formation of North America preserves an exceptional snapshot of early Eocene diversity in North America [32], but has only produced two aquatic taxa to date: *Presbyornis*, an anseriform, and *Limnofregata* Olson 1977 [33–35], likely an extinct relative of living frigatebirds [33,34]. Two semiaquatic taxa have also been recovered: an ibis-like taxon [36] and *Messelornis nearctica*, a stem ralloid [37,38]. Here we describe a new aquatic avian taxon from the early Eocene Fossil Butte Member (FBM, 51.97 ± 0.16 Ma [39]) of the Green River Formation. The taxon was originally illustrated and suggested to be anseriform in [32]. We recover the taxon as a member of Anseriformes. Most known Paleogene Anseriformes have a wide, duck-like beak (i.e. *Presbyornis*, *Nettapterornis* [40]), narrow pseudotoothed or dorsoventrally broad beaks (eg. *Pelagornis* [41] and *Gastornis*, respectively), or largely unknown beak morphology (i.e. *Conflicto antarcticus* [28]; *Vegavis iaai* [23,42,43]). The only other known exceptions to this are the latest Paleocene *Anachronornis anhimops* Houde et al. [44], the early Eocene *Danielsavis nazensis* Houde et al. [44], and the

Middle Eocene *Perplexicervix microcephalon* Mayr [45]. Like the latter taxa, the new fossil presents a narrow bill that is most similar to those of the Anhimidae but differs from all extant Anseriformes. We identify this fossil as the holotype specimen of a new species. New x-ray computed tomography (CT) images allow recovery of previously hidden morphologies, revealing it to represent a new taxon and ecology for the Green River Formation. The specimen was found at a near-shore locality of the Fossil Butte Member of the Formation, locality H [32,46], where several lithornithid and neoavian fossils have been previously described [32,47].

We additionally present a new morphological dataset for extinct and extant Galloanseres that includes more exemplars of extinct and extant Anseriformes than any previous matrix. Creation of this dataset included the examination and scoring of specimens of *Presbyornis* and *Telmabates antiquus* [48] that have not been included in previous studies. This included assessment of all *Presbyornis* material housed in the Vertebrate Paleontology Collection of the Smithsonian National Museum of Natural History and all *Telmabates* material housed in the Vertebrate Paleontology Collection of the American Museum of Natural History. We perform parsimony analysis on the new morphological matrix, Bayesian analysis on a matrix that combines the new morphological matrix with molecular data, and ancestral state reconstruction of ecological and behavioral traits. Results provide new insights into outstanding issues of avian phylogeny, evolution and diversification.

## Institutional abbreviations

AMNH, American Museum of Natural History, New York, NY, U.S.A.; FMNH, Field Museum of Natural History, Chicago, IL, U.S.A.; TMM, the Texas Memorial Museum, Austin, Texas, U.S.A.; USNM, National Museum of Natural History, Smithsonian Institution, Washington, D.C., U.S.A. Specimen numbers are presented in Table 1.

## Systematic paleontology

AVES Linnaeus, 1758 *sensu* [61]
 NEOGNATHAE Pycraft, 1900 *sensu* [61]
 ANSERIFORMES Wagler, 1831 [62]
 *Paakniwatavis grandei*, gen. et sp. nov.

### Holotype specimen

FMNH PA725, a partial skeleton and tracheal rings preserved in a kerogen-poor laminated micrite slab (Figs 1 and 2). Measurements are provided in Table 2. Most of the vertebrae are absent or obscured where present. The shoulder girdle, thoracic vertebrae, ribs, pelvis, femora, synsacrum, caudal vertebrae and pygostyle have been eroded due to taphonomic processes. It appears that bacteria-induced or some other organic erosion of the bone has occurred. This type and extent of organic erosion is unique within recovered avian fossils from FBM. Similar taphonomy has been reported in an early Cretaceous Enantiornithine [63], the early Cretaceous *Microraptor gui* [64], and several other Jurassic and Cretaceous avialan theropods [65,66], although much less erosion and deformation of the bone has occurred in these specimens compared to that of the holotype specimen of *P. grandei*. It has been suggested that this taphonomic phenomenon is due to changes in matrix chemistry caused by water being trapped between the feathers and the body [64]. Scanning electron microscopy and additional analysis of the holotype specimen of *P. grandei* is necessary to determine the cause of this rare taphonomy.

The holotype specimen was scanned using dual tube x-ray computed tomography at the PaleoCT Lab at the University of Chicago, which can scan specimens with a resolution of up to

**Table 1. Specimen numbers of skeletal specimens used for comparison during fossil description and phylogenetic analyses.**

| Group Name | Species Sampled and Specimen Numbers |
|---|---|
| Tinamiformes | *Crypturellus undulatus* (AMNH 2751, AMNH 6479), *Tinamus solitarius* (AMNH 21983, USNM 561269, USNM 345133) |
| Galliformes | *Lophura bulweri* (AMNH 10962, AMNH 16532, USNM 491472), *Gallus gallus* (AMNH 18555, AMNH 4031, M-12244, USNM 489422), *Crax alector* (USNM 621698), *Macrocephalon maleo* (USNM 225130), *Megapodius freycinet* (USNM 226175, USNM 557015) |
| Anseriformes | *Chauna torquata* (M-10449, USNM 646637, USNM 614549), *Anhima cornuta* (USNM 226166), *Anseranas semipalmata* (USNM 347638, USNM 621019), *Dendrocygna guttata* (USNM 560774), *Anas platyrhynchos* (USNM 633396, USNM 610643), *Anas platalea* (USNM 18549, USNM 614574), *Mergus serrator* (USNM 490105, USNM 634853, USNM 430710), *Chloephaga melanoptera* (USNM 491416, USNM 491887), *Amazonetta brasiliensis* (USNM 560068, USNM 635983), *Netta rufina* (USNM 292365), *Oxyura dominica* (USNM 430928), *Stictonetta naevosa* (USNM 612631), *Tadorna tadornoides* (USNM 638633), *Anser fabalis* (USNM 623342, USNM 643021), *Anser* (*Chen*) *caerulescens* (USNM 431549, USNM 345620), *Branta canadensis* (USNM 488584), *Cygnus atratus* (USNM 19738), *Coscoroba coscoroba* (USNM 346635) |
| Extinct Taxa | *Paakniwatavis grandei* (FMNH PA725), *Chaunoides* [49], *Gastornis* (*Gastornis giganteus* AMNH FR 70 [8,50]), *Pelagornis chilensis* [9], *Protodontopteryx ruthae* [51], *Telmabates antiquus* (AMNH FR 3166–3186 [48]), *Presbyornis* (*Presbyornis pervetus*: USNM PAL 483163–483166, USNM PAL 510082, USNM PAL 641330; *Presbyornis sp.*: USNM PAL ACC 2016973, USNM PAL ACC 335940, USNM PAL ACC 392324, USNM PAL ACC 393002, USNM PAL 498770–498771, USNM PAL 516605, USNM PAL 617185, USNM PAL 299845–299848, USNM PAL 618166–618180, USNM PAL 618183, USNM PAL 618189–618200, USNM PAL 618202, USNM PAL 618204–618207, USNM PAL 618209–618210, USNM PAL 618212–618215, USNM PAL 618218–618219, USNM PAL 618223–618224; *Presbyornis isoni*: USNM PAL 294117; [22,52]; CT scans from [53] of *Presbyornis sp.* USNM PAL 200846), *Vegavis iaai* [23,42], *Conflicto antarcticus* [28], *Nettapterornis oxfordi* [54], *Gallinuloides wyomingensis* [55], *Ichthyornis dispar* [56], *Lithornis promiscuus* (UNSM PAL 336570, USNM PAL 424072, USNM PAL ACC 378571, USNM PAL 391983; [57]; CT scans from [53] of *Lithornis promiscuus* USNM PAL 391983), *Calciavis grandei* [47], *Asteriornis maastrichtensis* [11], *Wilaru tedfordi* and *Wilaru prideauxi* [58,59], *Danielsavis nazensis* [44,60], *Anachronornis anhimops* [44,60], *Perplexicervix microcephalon* [45]. |

0.4 mm. As the specimen slab was large, it was scanned using a two-part multiscan that was combined to form one image sequence. The voxel size of the combined scan is 103.8710. The specimen is housed in the Department of Geology of FMNH. CT data generated during the current study are available in the Supplementary Data via DRYAD under DOI 10.5061/dryad.v15dv4208, and the rest of the Supplementary Data is available via Morphobank [67] under Project 4001 (http://morphobank.org/permalink/?P4001).

## Etymology

*Paakniwatavis* references *Paakniwat*, used by the Shoshoni tribe indigenous to the region of the recovery site and means "Water Spirit" [68]. The Water Spirits are dangerous supernatural beings that lure people to their death with child-like cries. The name references the aquatic ecology of this taxon. The species honors Dr. Lance Grande, who collected the holotype specimen, in recognition of his leading research on the faunas of the Green River Formation.

## Type locality and horizon

The holotype specimen was collected from FBM [69] Locality H (F-2 H in [32,45]). FBM Locality H is one of several near-shore localities that have produced avian fossils, and is located in the northeastern near-shore region. Locality H is within a four meter thick horizon representing a few hundred to a few thousand years of the early Eocene [46]. The horizons of the near-shore deposits of FBM are thicker than those of the mid-lake localities due to increased sedimentation near the shore. The fossil-bearing KPLM facies are characterized by thick

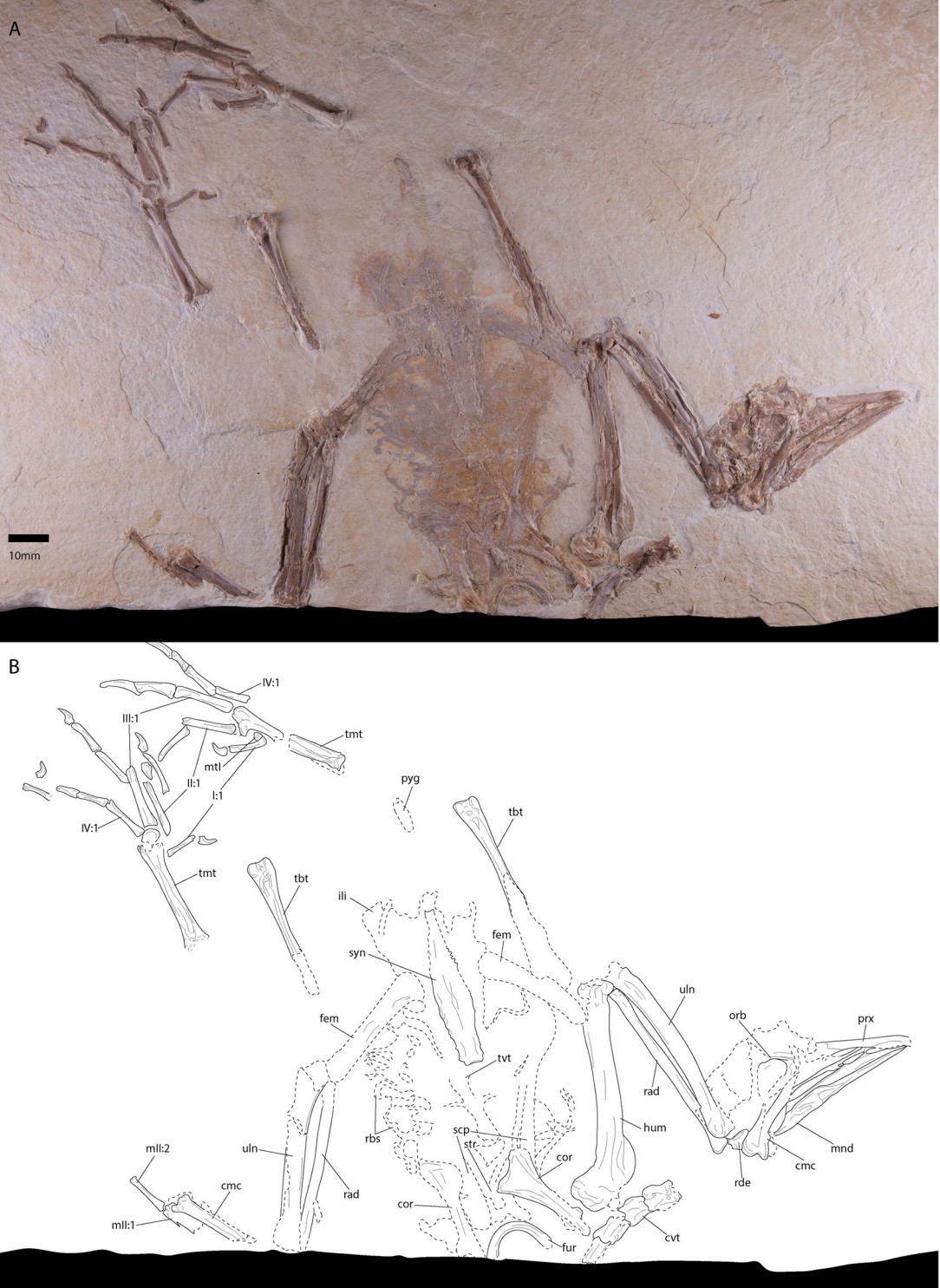

**Fig 1. Photograph (A) and line drawing (B) of the holotype specimen of *Paakniwatavis grandei* (FMNH PA725).** Extremely crushed bone and bone margin is delimited with dashed margins. Anatomical abbreviations: prx, premaxilla; orb, orbital margin; mnd, mandible; cvt, cervical vertebrae; tvt, thoracic vertebrae; syn, synsacrum; pyg, pygostyle; cor, coracoid; scp, scapula; fur, furcula; str, sternum; rbs, ribs; hum, humerus; uln, ulna; rad, radius; rde, radiale; cmc, carpometacarpus; mII:1, phalanx 1 of manual digit II; mtII:2, phalanx 2 of manual digit II; ili, ilium; fem, femur; tbt, tibiotarsus; tmt, tarsometatarsus; mtI, metatarsal I; I:1, phalanx 1 of pedal digit I; II:1, phalanx 1 of pedal digit II; III:1, phalanx 1 of pedal digit III; IV:1, phalanx 1 of pedal digit IV.

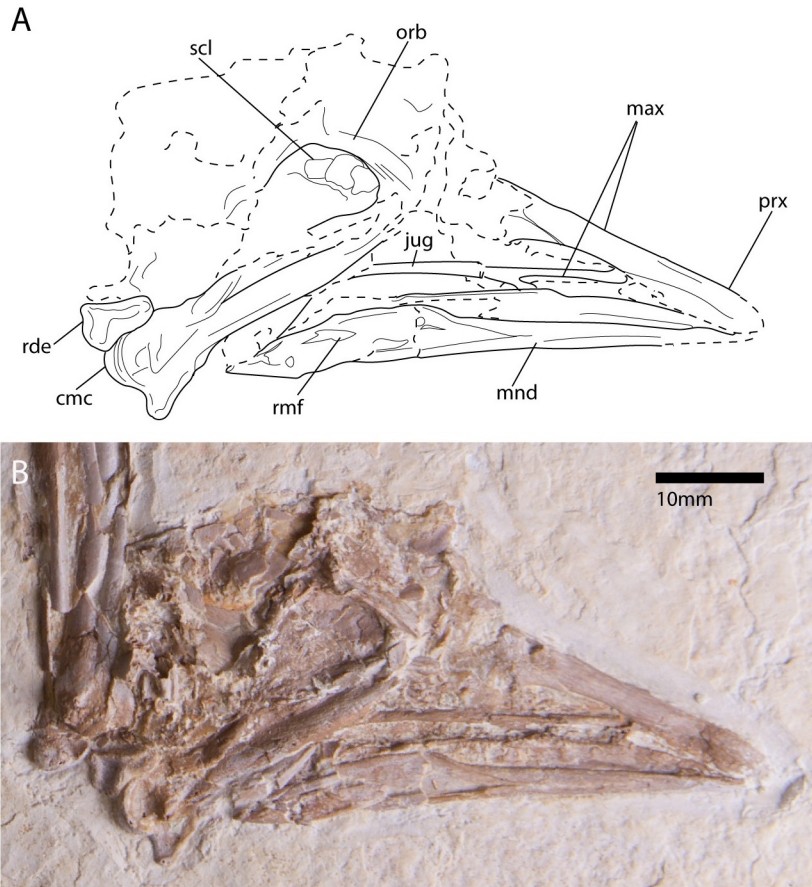

**Fig 2. Photograph (A) and line drawing (B) of the holotype specimen of *Paakniwatavis grandei* (FMNH PA725).**
Bone is unfilled. Extremely crushed bone and bone margin is delimited with dashed margins. Anatomical
abbreviations: prx, premaxilla; max, maxilla; jug, jugal; orb, orbital margin; rmf, rostral mandibular fenestra; scl, scleral
ossicles; mnd, mandible; rde, radiale; cmc, carpometacarpus.

kerogen-poor calcite laminae and instances of thin organic laminae. Laminae alterations can
be differentiated by inconsistent texture where the organic laminae is absent [46]. Locality H
has thus far yielded the highest number of avian fossils comprising lithornithids [47] (Palaeog-
nathae), *Gallinuloides wyomingensis* [37], see also galliform in [70], two frogmouth-like speci-
mens [71], a possible oilbird [35], four frigatebirds [33,34,72], an ibis-like taxon [36], a turaco
[38], two *Messelornis nearctica* specimens [70,73], a jacamar-like bird [70], a hoopoe-like bird
[32], stem rollers (Coraciiformes [74]), stem Psittacopasseres [75], additional taxa within Tell-
uraves [76,77], and several unknown birds [32]. The near-shore deposits are additionally char-
acterized by juvenile fish being more commonly preserved, abundant benthonic invertebrates,
stingrays (Batoidea), lizards, crocodiles, turtles, and non-flying mammals. Locality H also has
the only known amphibian preserved within FBM [78].

## Diagnosis

*Paakniwatavis grandei* is diagnosed by a proposed unique combination of characters compris-
ing (1) a mediolaterally narrow rostrum (character 3:state 1; Figs 1 and 3), (2) an elongate ret-
roarticular process (Fig 3; 241:2, 254:2), (3) a dorsoventrally thick furcula (Figs 1 and 3; 424:2),

**Table 2. Selected measurements of *Paakniwatavis grandei* in millimeters (mm), taken from surface of the holotype specimen slab (left/right) compared with taken and previously published measurements of *Presbyornis* [22,52], *Telmabates antiquus* [48], and *Nettapterornis oxfordi* [54].** Pedal phalanges are described using the format (digit:phalanx). Measurements are given for holotype specimens only.

| Measurement (mm) | *Paakniwatavis grandei* | *Presbyornis* | *Telmabates antiquus* | *Nettapterornis oxfordi* |
|---|---|---|---|---|
| **Total skull length** | 64.1 | 90 | | 100.0 |
| **Rostrum length** | 28.2 | 38.4 | | |
| **Furcula width** | 19.3 | | | 7.0 |
| **Coracoid length** | 32.7/~29.6 | 33.5–35.3 | 42.3–43.8 | 48.0 |
| **Coracoid width at midshaft** | 5.44/4.0 | | | 8.7 |
| **Humerus length** | -/71.3 | | | 119.3 |
| **Humerus width midshaft** | -/7.4 | | | 9.7 |
| **Ulna length** | 58.4/63.2 | | | |
| **Carpometacarpus length** | -/~30.2 | 46.5–50 | 63.1 | 69.5 |
| **Manual phalanx II: digit 1 length** | 15.5/- | 19.4 | 29.6–30.4 | 30.2 |
| **Manual phalanx II: digit 2 length** | 9.6/- | | | 22.7 |
| **Synsacral length** | 50.3 | | | >52 |
| **Pelvis acetabular width** | 19.5 | 5.1–6.4 | | |
| **Femur length** | 40.6/41.0 | 57–62 | 69–72 | |
| **Tibiotarsus length** | -/66.1 | | | |
| **Tarsometatarsus length** | 38.6/38.1 | 120 estimated, 105–120 estimated minimum | Minimum 105 estimated | |
| **I:1 length** | 9.5/11.1 | | | |
| **II:1 length** | 7.4/6.7 | | | |
| **II:2 length** | 13.4/14.3 | | | |
| **III:1 length** | 19.8/18.5 | | | |
| **III:2 length** | 13.7/13.1 | | | |
| **III:3 length** | 11.7/9.7 | | | |
| **IV:1 length** | 14.2/11.2 | | | |
| **IV:2 length** | 9.1/10.2 | | | |
| **IV:3 length** | 6.9/7.8 | | | |
| **IV:4 length** | 8.8/7.3 | | | |

(4) thoracic vertebrae that are not solely heterocoelous (286:2), (5) presence of a supracoracoid nerve foramen (Fig 4; 391:1), (6) lack of a spur on the carpometacarpus (509:1), (7) femora that are half the length of the tibiotarsi (Fig 1; 597:1), (8) presence of a prominent tubercle laterodistal to the pons supratendineus of the tibiotarsus (643:1), (9) tarsometatarsi that are just over half the length of the tibiotarsi (Fig 1; 656:1), (10) a medial hypotarsal crest that is projected farther plantar than the lateral crest (Fig 4; 668:1), and (11) a deep sulcus extensorius of the tarsometatarsus (Fig 1; 686:2). Diagnosis for the genus as per the species.

## Differential diagnosis

The rostrum shape exhibited by this taxon is unlike most other previously recovered Paleogene Anseriformes. Due to this *Paakniwatavis grandei* is easily distinguished from *Nettapterornis*, which has a mediolaterally wide, duck-like bill that is mediolaterally wider than the width measured from the left paroccipital process to the right paroccipital process. *Paakniwatavis* in contrast possesses a mediolaterally narrow bill that is narrower than the width between its paroccipital processes and, in this feature, is more similar to extant Anhimidae (Figs 1 and 3). The tomial margin of the bill of *Nettapterornis* is similarly dorsoventrally thick and recurved, unlike the straight and dorsoventrally narrow facial margin of the bill in *Paakniwatavis* (Fig 3).

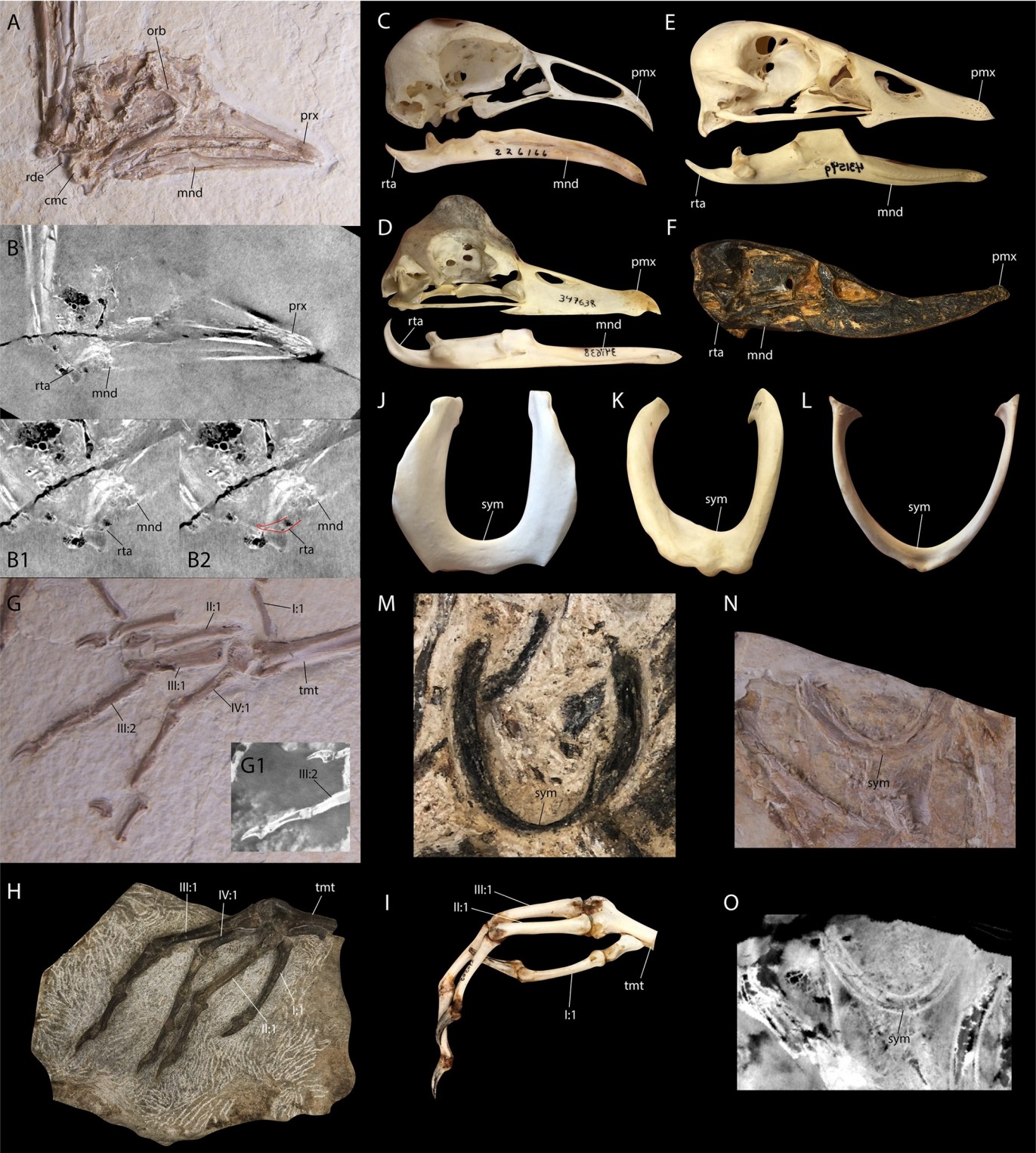

**Fig 3. Comparison of anseriform traits in *Paakniwatavis grandei* (FMNH PA725) to those of the extinct *Presbyornis* and extant Anseriformes (*Chauna 21orquate, Anseranas semipalmata,* and *Chloephaga melanoptera*).** Photograph (A) and CT scan slice (B) of the holotype specimen of *P. grandei* (FMNH PA725), with close-ups (B1 and B2) of CT scan slices of the caudal mandible. Photographs of the skull and mandible of (C) *C. 21orquate*, (D) *A. semipalmata*, I *C. melanoptera*, and (F) *Presbyornis sp.* (USNM 299846). Photograph (G) and CT scan slice (G1) of the left tarsometatarsus and pedal phalanges of *P. grandei*. Photographs of (H) the left tarsometatarsus and pedal phalanges of *Presbyornis sp.* (USNM ACC 335940) and (I) the right tarsometatarsus and pedal phalanges of

*21orquateata*. Photographs of the furcula of (J) *21orquateata*, (K) *A. semipalmata*, (L) *Anas platyrhynchos*, and (M) *Presbyornis sp.* (USNM ACC335940). Photograph (N) and CT scan slice (O) of the furcula of *P. grandei* from the holotype specimen. Anatomical abbreviations: prx, premaxilla; orb, orbital margin; mnd, mandible; rta, retroarticular process; rde, radiale; cmc, carpometacarpus; sym, symphysis; tmt, tarsometatarsus; I:1, phalanx 1 of pedal digit I; II:1, phalanx 1 of pedal digit II; III:1, phalanx 1 of pedal digit III; III:2, phalanx 2 of pedal digit III; IV:1, phalanx 1 of pedal digit IV.

The nares of *Paakniwatavis* are over half the length of the rostrum, whereas that of *Nettapterornis* is less than half of their rostral length. The coracoid in *Paakniwatavis* is more elongate relative to the width of the sternal facet than that of *Nettapterornis* (Fig 4). The rami of the furcula are extremely thick in *Paakniwatavis* (Fig 1) compared to the thin furcula of *Nettapterornis* (Fig 3). The sulcus ligamentosus transversus is more truncate in *Paakniwatavis* than in *Nettapterornis*. The fossa olecrani of *Paakniwatavis* is more shallow than that of *Nettapterornis*. *Paakniwatavis* has a deeper fossa infratrochlearis of the carpometacarpus than *Nettapterornis*. The craniocaudal lengths of manual digits II and III at the synostosis are subequal in *Paakniwatavis*, whereas II extends further distally than III in *Nettapterornis*.

As with *Nettapterornis*, *Paakniwatavis* can be differentiated from *Presbyornis* due to *Presbyornis* possessing a mediolaterally wide, duck-like bill that is mediolaterally wider than the width measured from the left paroccipital process to the right paroccipital process. The tomial margin of the bill of *Presbyornis* is again like that of *Nettapterornis* and dorsoventrally thick and recurved, unlike the straight and dorsoventrally narrow facial margin of the bill in

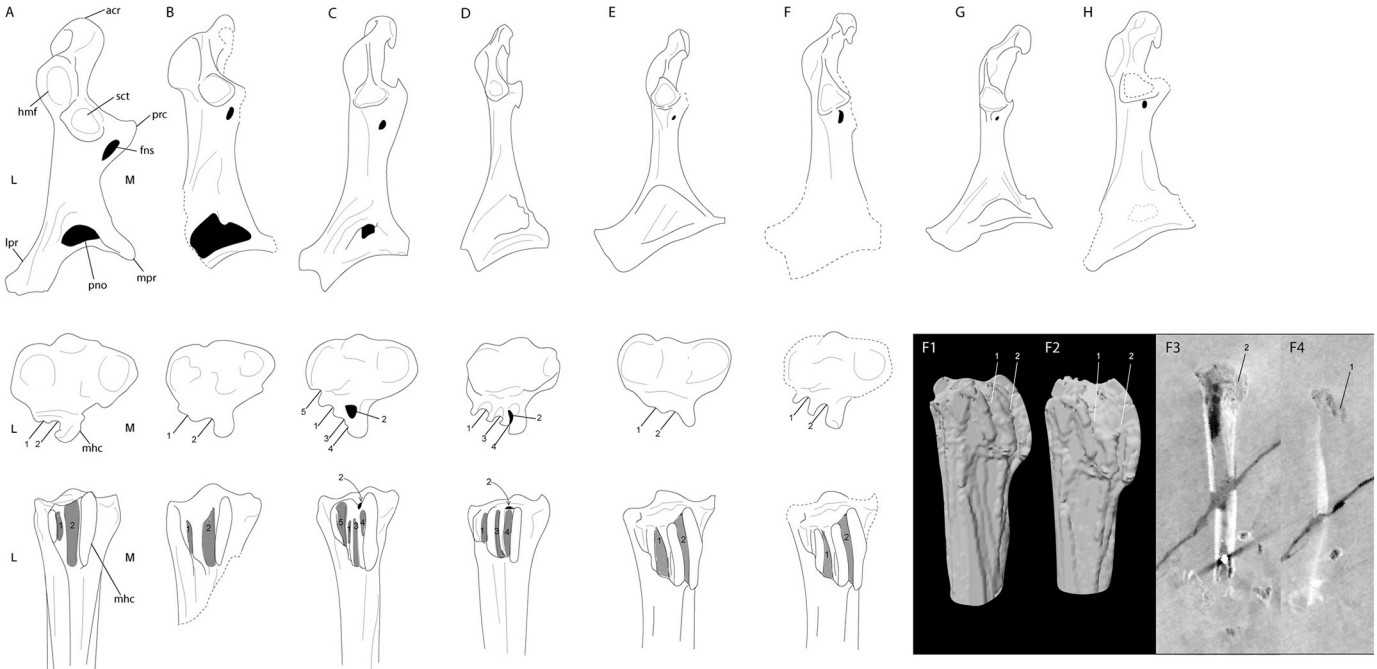

**Fig 4. Comparison of the coracoid and hypotarsus of the tarsometatarsus of *Paakniwatavis grandei* (FMNH PA725) to that of the extinct *Presbyornis*, *Telmabates antiquus*, *Chaunoides antiquus* and *Nettapterornis oxfordi* and extant Anseriformes.** Line drawings of the coracoids and hypotarsi of (A) *Anhima cornuta*, (B) *C. antiquus*, (C) *Anseranas semipalmata*, (D) *Dendrocygna guttata*, (E) *Presbyornis sp.*, (F) *P. grandei*, (G) *T. antiquus* and (H) *A. oxfordi*. Coracoids are depicted in dorsal aspect, hypotarsi are depicted in proximal and plantar aspects. L and M in panel (A) denote the lateral and medial sides of each element. (F1 and F2) are segmented hypotarsi of the left tarsometatarsus of *P. grandei* in lateroplantar aspect. (F3 and F4) are CT scan slices of the same tarsometatarsus of *P. grandei*. Anatomical abbreviations: acr, processus acrocoracoideus; hmf, humeral facet; sct, scapular facet; prc, procoracoid process; fns, supracoracoid nerve foramen; pno, pneumatic opening; lpr, lateral process; mpr, medial process; mhc, medial hypotarsal crest; 1, sulcus for tendon of musculus flexor hallucis longus (fhl); 2, sulcus or canal for tendon of musculus flexor digitorum longus (fdl); 3, sulcus for tendon of musculus flexor perforatus digiti 2 (fp2); 4, sulcus for tendon of musculus flexor perforans et perforatus digiti 2 (fpp2); 5, sulcus for muscularus fibularis longus (fbl).

*Paakniwatavis* (Fig 3). Similarly, the nares of *Paakniwatavis* are over half the length of the rostrum, whereas those of *Presbyornis* are less than half of their rostral length. The crista tympanica of the quadrate terminates within the ventral half of the quadrate in *Paakniwatavis*, whereas it terminates within the dorsal half in *Presbyornis*. The tuberculum subcapitulare is separated from the squamosal capitulum in *Paakniwatavis*, but is contiguous with the capitulum in *Presbyornis*. The relative heights of the rostral and caudal apices of the coronoid process of the mandible are subequal in *Paakniwatavis*, but the rostral apex is higher in *Presbyornis*. The mandibular ramus caudal to the rostral fenestra mandibulae is deep and concave along the medial face in *Paakniwatavis*, and is relatively shallow in *Presbyornis*. A mandibular ventral angle is prominent in *Presbyornis*, but not in *Paakniwatavis*. While the fenestra rostralis mandibulae is slit-like in *Paakniwatavis*, it is transverse and largely perforate in *Presbyornis* (Fig 3). The synsacral count of *Paakniwatavis* is within 14–19 vertebrae, whereas it is within 10–13 for *Presbyornis*. In the coracoid, a small pneumatic foramen directly below the scapular cotyla is present in *Presbyornis* but is absent in *Paakniwatavis* and *Nettapterornis*. The rami of the furcula are again extremely thick in *Paakniwatavis* compared to the thin furcula of *Presbyornis*. The acromion process of the scapula is truncate unlike the cranially elongate processes of *Presbyornis*. The dorsal angle of the scapula is caudal to the midpoint of the shaft in *Presbyornis*, but is at the midpoint in *Paakniwatavis*. The fossa pneumotricipitalis of the humerus is pneumatic in *Paakniwatavis* and *Nettapterornis* but apneumatic in *Presbyornis*. *Paakniwatavis* has a narrower crista deltopectoralis of the humerus than *Presbyornis*. *Paakniwatavis* has a deeper impressio coracobrachialis cranialis than *Presbyornis*, The sulcus ligamentosus transversus is more truncate in *Paakniwatavis* than in *Presbyornis*. The fossa m. brachialis is located more medially in *Paakniwatavis*. The fossa olecrani of *Paakniwatavis* is again more shallow than that of *Presbyornis*. The depression radialis of the ulna is deeper in *Paakniwatavis*. The labrum dorsalis of the carpometacarpus is more sharply angled in *Paakniwatavis*. The epicondylus medialis of the tibiotarsus is more pronounced in *Paakniwatavis* than in *Presbyornis*. The epicondylus medialis is less pronounced than those of *Presbyornis* or *Chaunoides antiquus*. The sulcus extensorius tibiotarsus opens under the pons supratendineus along the midline in *Paakniwatavis* rather than medially. The tarsometatarsus is approximately half the length of the tibiotarsus or less in *Paakniwatavis*, whereas the length of these elements is subequal in *Presbyornis*. The lateral cotyle of the tarsometatarsus is more shallow in *Paakniwatavis*. In *Paakniwatavis*, pedal digit IV: phalanx IV is longer than IV: III, whereas the opposite is true in *Presbyornis*.

When compared to extinct taxa, the narrow bill of *Paakniwatavis* is most like those of *Anachronornis anhimops* and *Danielsavis nazensis*. The rostrum of *Paakniwatavis* can be distinguished from that of *Danielsavis* as the nares of *Paakniwatavis* do not extend as far anteriorly into the rostrum as those of *Danielsavis* do. The coronoid processes of the mandible are tuberculate and subtle compared to those of *Danielsavis* (Fig 2, Mayr et al. [60]). The rami of the furcula are much thicker in *Paakniwatavis* compared to the thin furcula of of *Danielsavis*. The acromion process of the scapula is truncate and unlike the cranially elongate processof *Danielsavis*. The acrocoracoid process of the coracoid is more sharply hooked than that of *Danielsavis* (Fig 5, Mayr et al. [60]). The shaft of the coracoid of *Paakniwatavis* is wider and more robust compared to that of *Danielsavis*. The impressio m. sternocorocoidei is much deeper in *Paakniwatavis* compared to the shallow impression in *Danielsavis* (Fig 4). A robust additional medial projection above the medial angle that is part of the origin of the sternocoracoclavicular ligament is present in *Danielsavis* but absent in *Paakniwatavis*. The length of the ulna is longer than that of the humerus in *Danielsavis*, whereas in *Paakniwatavis* the ulna is slightly more truncate than the humerus. *Danielsavis* has a robust index process that extends beyond the articular facet of manual digit II:1 which is not present in *Paakniwatavis*.

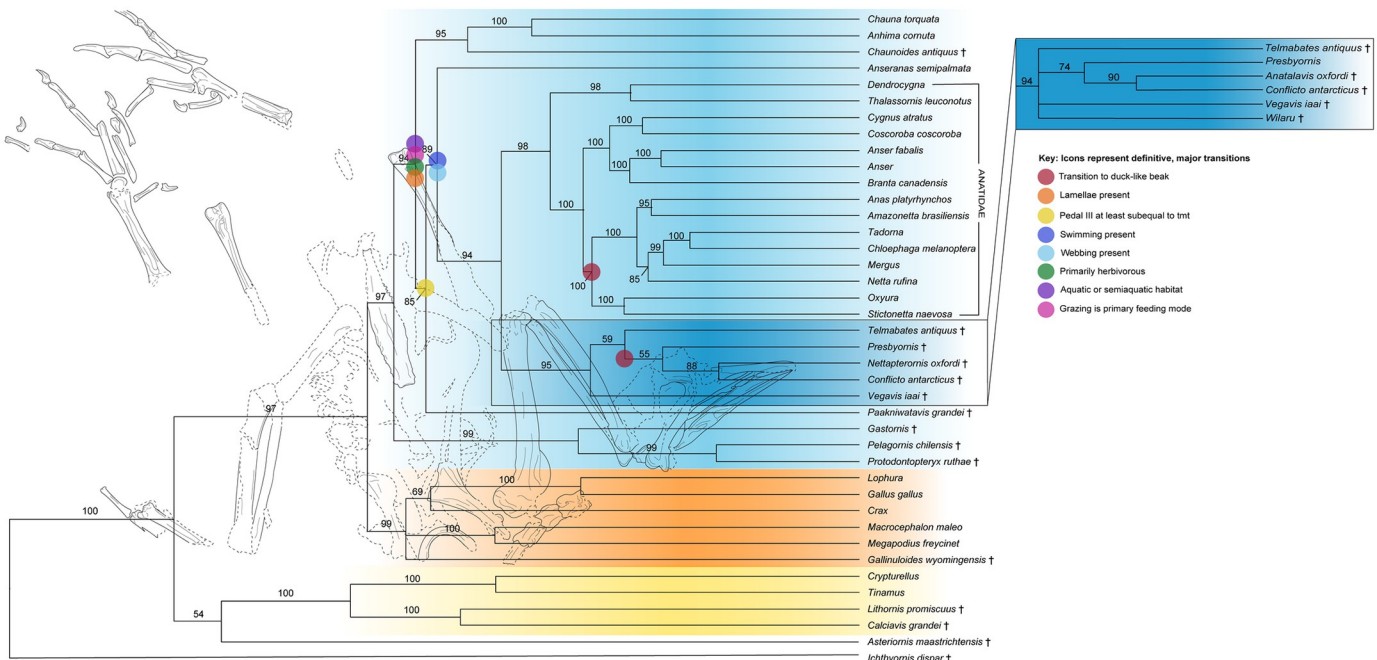

**Fig 5. Resulting consensus tree from Bayesian analysis of 719 morphological characters and 158,368 base pairs.** Clade credibility values greater than 50% are annotated above branches. Extinct taxa are delimited with daggers. A line drawing of the holotype specimen of *Paakniwatavis grandei* is overlaid on the tree and a reconstruction of this species is shown to the right of the tree. Icons represent definitive ancestral state reconstruction of the earliest transitions. Only the first major transitions for Anseriformes are shown, with subsequent transitions excluded. The insert to the right of the tree displays how results changed when *Wilaru* was included in analysis.

The angulus mandibulae of the mandible is more truncate in height compared to that of *Anachronornis* (Fig 2, Mayr et al. [60]). The rami of the furcula are much thicker in *Paakniwatavis* compared to the thin furcula of *Anachronornis*. The shaft of the coracoid of *Paakniwatavis* is again wider and more robust compared to that of *Anachronornis*. The impressio m. sternocorocoidei is much further excavated in *Paakniwatavis* compared to the shallow impression in *Anachronornis* (Fig 4). A robust additional medial projection above the medial angle is again present in *Anachronornis* but absent in *Paakniwatavis*. *Paakniwatavis* has a deeper impressio coracobrachialis cranialis than *Anachronornis*. The sulcus ligamentosus transversus of the humerus is deeper in *Paakniwatavis* than in *Anachronornis*. The head of the humerus in *Anachronornis* is triangular unlike the rounded head in *Paakniwatavis* and is projected much further proximally than that of *Paakniwatavis*.

As in *Anachronornis* and *Danielsavis*, *Paakniwatavis* can be differentiated from *Perplexicervix* due to the shaft of the coracoid of *Paakniwatavis* being wider and more robust in *Paakniwatavis*, and again *Paakniwatavis* possesses a more deeply excavated impressio m. sternocorocoidei (Fig 4). Additionally, the supracoracoid nerve foramen of *Perplexicervix* is smaller and located along the midline of the shaft, whereas it is larger and located further medially in *Paakniwatavis*. *Paakniwatavis* can be distinguished from *Telmabates* due to the number of synsacral vertebrae present. The synsacral count of *Paakniwatavis* is within 14–19 vertebrae, whereas it is within 10–13 for *Telmabates* as it is in *Presbyornis*. *Telmabates* has a small pneumatic foramen directly below the scapular coracoid like *Presbyornis*, and again this is absent in *Paakniwatavis*. The acromion process of the scapula is again truncate and unlike the cranially elongate processes of *Telmabates*. The dorsal angle of the scapula is caudal to the

midpoint of the shaft in *Telmabates*, but is at the midpoint in *Paakniwatavis*. The incisura capitis of the humerus is deep in *Paakniwatavis* but shallow in *Telmabates*. The fossa pneumotricipitalis of the humerus is pneumatic in *Paakniwatavis* but apneumatic in *Telmabates*. *Paakniwatavis* has a deeper impressio coracobrachialis cranialis than *Telmabates*, The sulcus ligamentosus transversus of the humerus is more truncate in *Paakniwatavis* than in *Telmabates*. The epicondylus medialis of the tibiotarsus is more pronounced in *Paakniwatavis* than in *Telmabates*. *Paakniwatavis* has a dome-like crista that overhangs the proximal margin of the fossa pneumotricipitalis of the humerus, whereas this crista is typical in *Telmabates*. The impressio m. pectoralis is deeper in *Paakniwatavis*. The trochlea carpalis of the carpometacarpus is deeper in *Paakniwatavis*. The epicondylaris medialis depression is more shallow in *Paakniwatavis* than in *Telmabates*.

*Paakniwatavis* can be differentiated from *Chaunoides* Alvarenga 1999 due to differences in the coracoid and tarsometatarsus. Within the coracoid, the primary axis of the scapular cotyla is skewed laterally in *Paakniwatavis*, *Presbyornis*, *Telmabates* and *Nettapterornis* but is centralized in *Chaunoides* (Fig 4). The sternal facet is skewed ventrally in *Chaunoides*, whereas it projects caudally in *Paakniwatavis*. In the tarsometatarsus, the major hypotarsal ridge in *Paakniwatavis* is hooked distally. The epicondylus medialis of the tibiotarsus is more pronounced in *Paakniwatavis* than in *Chaunoides*.

## Description and comparison

**Skull and mandible.** The skull and rostrum are preserved in dorsolateral aspect. The right carpometacarpus has broken through the skull and mandible and caused deformation along the caudoventral margins of these elements; however, many of the elements obscured by the carpometacarpus can still be seen in the CT scan. The bill length is roughly equal to that of the cranium. It is mediolaterally narrow and tapered toward the anterior margin, as in Anhimidae, but the terminus of the rostrum is not strongly hooked like those of Anhimidae. It is only slightly decurved and is most similar to those of Anseranatidae. The tip of the premaxilla additionally appears to be slightly thickened, as in Anseranatidae. This is unlike the mediolaterally broad, duck-like bills of most anseriform-like fossils such as *Nettapterornis* and *Presbyornis* and most like those of *Anachronornis* and *Danielsavis* [44,60]. The caudal and anterior portions of the nares are largely broken, but CT scans (see Supplementary Data via 10.5061/dryad.v15dv4208) show that the nares would have been holorhinal and rostral to the zona flexoria craniofacialis (ZFC), as in all Galloanserines. The length of the nares is over half that of the rostrum, a condition only present in Anseriformes within Anhimidae. The cranium immediately caudal to the ZFC appears to have had the pneumatized swelling seen in *Chauna* which is described in [13] (character 10). The overall shape and size of the cranium is most similar to those of Anhimidae or Anseranatidae. The robust left jugal is present between the rostrum and left ramus of the mandible. A prominent, lateroventrally projecting supraorbital crest is present, like those of Anhimidae [79]. CT data reveals that the postorbital process is elongate, as in most Anseriformes (Supplementary Data via 10.5061/dryad.v15dv4208). The zygomatic process is absent, as in all Anseriformes, including both Anhimidae and Anseranatidae. At least three large, broad scleral ossicles have been preserved along the rostral margin of the orbit. Fonticuli within the interorbital area appear to be absent.

The mandible is dorsoventrally thin along the proximal margin and widens caudal to the coronoid process. While both rami are visible on the surface of the slab, much of the left mandible and the left quadrate are obscured by the carpometacarpus and are only visible within the CT data (Supplementary Data via 10.5061/dryad.v15dv4208). The rostral terminus of the mandible is broken and it cannot be assessed whether it was decurved, as in Anhimidae. The

rostral mandibular fenestrae are rostrocaudally elongate and slit-like but appear to open rostrally. The shape, angle and placement of the fenestrae are most similar to those of Anseranatidae. The coronoid processes are tuberculate and subtle compared to those of most Anseriformes, including those of Anhimidae and *Danielsavis* (Fig 2, Mayr et al. [60]). Similarly, the angulus mandibulae are truncate in height compared to those of many Anseriformes, including *Anachronornis* (Fig 2, Mayr et al. [60]). CT data reveals an elongate medial process at the condylar area of the mandible, as in all Galloanserines. On the right ramus, a slender, recurved retroarticular process that has been severed from the mandible by the carpometacarpus is visible in the CT data (Fig 3, Supplementary Data via via 10.5061/dryad.v15dv4208). It is significantly dorsoventrally thinner than that of *Anachronornis* (Fig 2, Mayr et al. [60]). A retroarticular process is present in all Galloanserines and most included fossils (Fig 3).

The left quadrate is visible in the CT data (Supplementary Data via 10.5061/dryad.v15dv4208). Most features of the quadrate have been obliterated, including the orbital crest. No foramen is present between the capitulae, but a foramen is present on the medial face of the otic process, as in all Anseriformes. A tuberculum subcapitulare is present as in all Galloanserines. A prominentia submeatica appears to be present, as in all Anseriformes.

**Axial skeleton.**   The atlas, axis and the third cervical vertebra are poorly preserved in ventrolateral aspect near the furcula, with two additional cervical vertebrae preserved caudal to the third cervical vertebra in ventral aspect. The dorsal spines are rounded and not dorsoventrally prominent. This condition is more similar to that of Anseranatidae rather than that of Anhimidae. The third cervical vertebra appears to be more elongate and to have a less prominent dorsal spine, suggesting that the cervical series elongates caudally.

The thoracic and pelvic areas of the specimen are poorly preserved. The bone appears to have eroded due to taphonomic processes, possibly due to bacterial erosion. Several caudalmost cervical vertebrae and thoracic vertebrae are visible on the surface of the slab and within the CT data (see Supplementary Data via 10.5061/dryad.v15dv4208). Most of the thoracic vertebrae are preserved in ventral aspect. CT data reveals that the thoracic series is not completely heterocoelous. This condition is present in *Presbyornis*, *Nettapterornis* and *Telmabates* but lost in extant Anseriformes. The thoracic vertebrae do not fuse to form a notarium. This condition is present in all extant Anseriformes with the exception of Anseranatidae.

The synsacrum is preserved in ventral aspect. CT data reveals that at least 14 synsacral vertebrae are present (see Supplementary Data via 10.5061/dryad.v15dv4208). This is the condition in all extant Anseriformes. *Telmabates* and *Presbyornis* have 13 or fewer synsacral vertebrae. The sulcus ventralis of the synsacrum is present and appears to have been deep. A deep sulcus ventralis is also present in both Anhimidae and Anseranatidae. Several poorly preserved caudal vertebrae with indiscernible features are present in the CT data (see Supplementary Data via 10.5061/dryad.v15dv4208). A craniocaudally stout pygostyle is visible caudal to the synsacrum on the surface of the slab.

**Shoulder girdle.**   The symphysis and ventral portion of the clavicles of the furcula are preserved in caudal aspect. The furcula is more robust than those of most Anatidae but more gracile and thin than those of Anhimidae or Anseranatidae (Fig 3). A processus interclavicularis dorsalis is absent, and an apophysis is absent.

Both coracoids are preserved in ventral aspect. The sternal bases and shafts of the coracoids can be seen. The acrocoracoid process is robust and hooked, like that of *Anachronornis* (Fig 5, Mayr et al. [60]). CT data reveals a procoracoid process and supracoracoid nerve foramen to be present. A procoracoid process is present in many extant and extinct Anseriformes, including Anhimidae and Anseranatidae (Fig 4). Supracoracoid nerve foramina are present in all included extinct Anseriformes with the exception of *Conflicto*, Anhimidae, and Anseranatidae. They are also present within the included Pelagornithidae. Supracoracoid nerve foramina are

absent in all included extant Anatidae with the exception of *Cygnus atratus*. Portions of the scapulae are present near to or overlapping the coracoids, but most of their morphology is indiscernible.

**Forelimbs.** The right humerus is preserved in cranioventral aspect. The sulcus ligamentosus transversus is extremely deep; more like the condition in Anseranatidae than that of Anhimidae. The crista deltopectoralis is prominent, rounded and flares cranio-laterally like those of most Anseriformes. The caput humeri is bulbous and prominent. Also, as in *Anas*, the impressio coracobrachialis is relatively deep. The condylus dorsalis is small and hamate. The tuberculum supracondylare ventrale is large and bulbous like that of *Anas*. The epicondylus dorsalis appears to be distally extensive, and most similar to that of *Anas*. CT data reveals the morphology of the caudal humerus; the crista proximally edging the fossa pneumotricipitalis appears to have been domed slightly, like that of *Nettapterornis*.

The right radius and ulna are preserved in ventral aspect. Most features of these bones cannot be seen or were not preserved. The ulnar body is thick and robust, but shorter than that of the humerus. The olecranon is much shorter and more rounded than that of *Anas*, and is most similar in size and shape to that of *Chauna*. The impressio brachialis appears to be proximally deep and ovoid, similar those of Anhimidae. Much of the distal portion of the ulna is obscured by the skull. There appears to be small pneumatic foramen just under the cotyla humeralis of the radius. The left ulna and radius are present but are not as well preserved as those of the right. They are preserved in dorsal aspect.

The right carpometacarpus overlaps the mandible and has broken on top of the skull. It is preserved in ventral aspect. The processus pisiformis is elongate, rounded and caudally oriented; it is most similar to that of Anseranatidae. The rim of the dorsal trochlea is prominent and strongly angled, which is also similar to the condition of Anseranatidae. The processus extensorius is identical in size and shape to that of Anseranatidae as it is triangular in shape with a rounded point. The os carpi ulnare is present as well and is visible in the CT scans (see Supplementary Data via 10.5061/dryad.v15dv4208). The os metacarpale minus is obscured by the skull but appears to be dorso-ventrally thick. Only the distal shaft and condylar area of the left carpometacarpus is preserved and is in cranial aspect. The facies articularis digitalis major on the left carpometacarpus form a 3-pronged distal end with rounded termini. A prominent crista is present towards the dorsal aspect of the digital articular area; it is not seen in *Anas* and in *Chauna* it is hook-like. The os metacarpale minus appears broken and only a small spatium intermetacarpale appears present, but this could be exaggerated by crushing. The phalanges digitus majoris are preserved in dorsal aspect, although digit I is somewhat obscured by the carpometacarpus. The pila cranialis of phalanx dig. majoris I appears robust. Phalanx II of this digit is thin and elongate.

**Sternum.** The sternum is extremely poorly preserved but appears to have been broad and subrectangular like that of *Anas*. A prominent carina is distorted but preserved. An elongate spina externa appears to be present, but it cannot be discerned with confidence whether this is part of the sternum or a vertebra lying under the sternum. What appears to be an isolated uncinate process looks visible on a rib located to the left of the sternum; however, this also cannot be assessed with confidence.

**Pelvic girdle.** The pelvis is preserved in ventral aspect. A pair of acetabular struts are visible in the CT data. Portions of the postacetabular ilium, ischium and a medially curved pubis can be seen more clearly in the CT data.

**Hindlimbs.** The right and left femora are poorly preserved. The head of the right femur appears robust like those of Anhimidae or Anseranatidae. The femora are half the length of the tibiotarsi. The right and left tibiotarsi are preserved in cranial view. All that remains of the left tibiotarsus is the mid and distal shaft and the condylar area. The bodies of the tibiotarsi are

long and slender. The entire right tibiotarsus appears is preserved along with the head and proximal shaft of the right fibula. The cranial cnemial crest is obscured by the right femur, but appears prominent and acuminate, very similar to that of *Anseranas*. The canalis and sulcus extensorius appear to be deep, and the pons supratendineus is ossified. No intratendinous ossification is present.

The tarsometatarsi are just over half the length of the tibiotarsi and are preserved in mediodorsal aspect. The hypotarsus is visible in the CT data (Fig 4). The medial hypotarsal crest is more prominent, as in most Anseriformes. Two sulci for flexor tendons are present. A canal for the flexor digitorum longus (FDL) tendon is not present. This condition is most similar to those of Anhimidae, *Chaunoides*, and *Presbyornis*, although the hypotarsal sulci of *P. grandei* are much deeper and are more comparable in depth to those of the anatid *Dendrocygna guttata* (Fig 4). Anseranatidae conversely have an FDL tendon that is enclosed in a canal and exhibit four shallow sulci for flexor tendons (Fig 4 and [80]). The hypotarsi of all extant Anatidae are trisulcate, with a canal present for the FDL tendon, as in *Dendrocygna* (Fig 4 and [80]). Both tarsometatarsi exhibit deep extensor sulci that extend to the distal portion of the tibiotarsus. In the right tarsometatarsus, trochlea metatarsal III reaches the most distally. Trochlea metatarsal II is deflected plantarly, as in Anatidae. The phalanges are elongate like those of *Anas*. IV is shorter than III but the hallux is elongate. Phalanx III is longer than the length of the tarsometatarsus.

**Body mass.** Femur length was the best predictor of body mass in extant volant birds ($R^2$ = 0.9028 [81]) that could be obtained from FMNH PA725, *Presbyornis* [22,52] and *Telmabates* [48]. Average femur length measurements were used to calculate body mass for each taxon. An approximate mean body mass estimate for *P. grandei* is 304g based on the published allometric equation using femoral length from [81]. The approximate mean body mass estimates for *Presbyornis* and *Telmabates* are 882g and 1423g, respectively based on the same equation.

## Materials and methods

### Nomenclatural acts

The electronic edition of this article conforms to the requirements of the amended International Code of Zoological Nomenclature, and hence the new names contained herein are available under that Code from the electronic edition of this article. This published work and the nomenclatural acts it contains have been registered in ZooBank, the online registration system for the ICZN. The ZooBank LSIDs (Life Science Identifiers) can be resolved and the associated information viewed through any standard web browser by appending the LSID to the prefix ""http://zoobank.org/"". The LSID for this publication is: urn:lsid:zoobank.org:pub:2D9C382C-87C9-49DB-929D-E99D3871F316. The electronic edition of this work was published in a journal with an ISSN, and has been archived and is available from the following digital repositories: PubMed Central, LOCKSS, bioRxiv.

### Phylogenetic analyses and ancestral state estimation

**Comparative materials.** Specimens used for the description and phylogenetic analyses came from the Bird Division of FMNH, the Ornithology Department of AMNH, the Ornithology Department of USNM, and the Texas Memorial Museum. Osteological terminology largely follows [82]. Specimen numbers for examined taxa are presented in Table 1. Extinct taxa were scored from direct observation where possible. CT scans of the skulls of *Presbyornis* and *Lithornis promiscuus* were loaned from USNM for examination (originally produced for [53]). All USNM *Presbyornis* material and all AMNH *Telmabates antiquus* material was

directly examined. Extinct taxa scored from photographs when necessary comprise *Ichthyornis dispar* [56,83], *Gastornis gigantea* and *Gastornis steini* [7,50], *Pelagornis chilensis* [9], *Protodontopteryx ruthae* [51], *Vegavis iaai* [23,42,43], *Conflicto antarcticus* [28], *Nettapterornis oxfordi* [54], *Wilaru tedfordi* [59] and *Wilaru prideauxi* [58]), *Presbyornis* [22,52], *Chaunoides antiquus* [49], *Anachronornis anhimops* [44], *Danielsavis nazensis* [44], *Perplexicervix microcephalon* [45], *Lithornis promiscuus* [57], *Calciavis grandei* [71], *Asteriornis maastrichtensis* [11], *Gallinuloides wyomingensis* [55], *Danielsavis nazensis* [44,60], *Anachronornis anhimops* [44,60], and *Perplexicervix microcephalon* [45]. Scorings for separate *Gastornis*, *Presbyornis* and *Wilaru* species were concatenated into genus-level taxa for more robust phylogenetic placement, especially since most recovered *Presbyornis* specimens have not been assigned to the species level.

**CT scan.** PA 725 was CT scanned at the University of Chicago by April Neander, using the facilities of the Department of Organismal Biology and Anatomy. A two-part multiscan was conducted in order to achieve the highest resolution possible. The scan sheet is available on Morphobank [67] under Project 4001 (http://morphobank.org/permalink/?P4001) and the scan sheet along with all CT tiffs are available on DRYAD under DOI 10.5061/dryad.v15dv4208. The multi scan was reconstructed with the aid of Matthew Colbert at the University of Texas at Austin. CT scans were segmented by the authors at the University of Texas at Austin using the Avizo program. All segmentation files are available on Morphobank.

**Character matrices and ancestral state reconstruction.** The morphological data matrix is built on that of [79,84] following the methodology discussed in those publications, but was modified to comprise 719 discrete characters and 41 taxa, 16 of which are extinct. *Anachronornis*, *Danielsavis* and *Perplexicervix* were later added to the matrix following release of the recent publications on these taxa [44,45,60]; however, addition of these taxa did not change the phylogenetic results of any analyses. Due to this, we have included discussion of their relationships within the manuscript but have made the matrix and resulting trees that include these taxa publicly available in the Supplementary Data on Morphobank [67] under Project 4001 (http://morphobank.org/permalink/?P4001).

Most of the characters detail skeletal morphology, although several characters additionally describe musculature, the syrinx, behavior, and ecology. Ten additional characters were added to the morphological matrix for ancestral state reconstruction of relevant traits that detail habitat preference, swimming mode, diet, status of rhamphothecal lamellae, status of pedal webbing, feeding mode, syrinx anatomy, and the relative length of pedal phalanx III compared to that of the tarsometatarsus. Ancestral state reconstructions were performed in Mesquite [85] using parsimony methods. All character descriptions are provided in the S1 Appendix and all data matrices and analyses logs have been made publicly available on Morphobank [67].

Characters from several previously published large-scale morphological datasets focused on early avian divergences and galloanserine-like fossils [13,14,27,52,86–89] were evaluated for use in this iteration of the Musser and Cracraft [79] dataset, and characters from these matrices have been cited where a character was incorporated from a previously published matrix or where characters overlap with our matrix. Characters were additionally used from several previously published matrices which have also been cited in the S1 Appendix.

We additionally created a combined data matrix comprising the morphological data coupled with all available mitochondrial genomes available on GenBank [31,90] for this taxon sampling and the Early Bird II dataset from [91]. The combined dataset has a total of 158,368 base pairs. This data matrix is also publicly available under the same project on Morphobank [67]. Sequences included were matched to the species level where possible; otherwise sequences of taxa within the same genus or family level were used. Mitochondrial genomes were aligned using MAAFT [92]. The mitochondrial genome data was not partitioned.

Jmodeltest [93] was used to find the best fit substitution model, GTR+I+G as is common in vertebrate nonpartitioned mitochondrial genome data [94]. We used the GTR + G model for the Early Bird II dataset, as in [95], but did not partition the dataset due to using a limited taxon sample.

**Phylogenetic analyses.**   We performed unconstrained heuristic parsimony analyses of the morphological dataset in PAUP* [95] Version 4.0a, build 169 using 10,000 random taxon addition replicates per run. A molecular backbone constraint that minimally constrained Galloanseres to be monophyletic needed to be employed when *Wilaru* was included, as inclusion of this taxon placed Galliformes as the sister taxon of included Palaeognathae. Heuristic search algorithms were used. Tree bisection reconnection branch swapping was employed and minimum branch lengths valued at zero were collapsed, following Mayr and Clarke [88] and Musser and Cracraft [79]. No character weighting was applied. All characters were unordered. Bootstrap analyses were performed using 500 bootstrap replicates each with 10 random taxon addition replicates, as in Mayr and Clarke [88].

Within the combined data analysis, mitochondrial genomes were analyzed using a GTR + invariable gamma substitution model [96,97]. The Early Bird II dataset was analyzed using a GTR + gamma substitution model. Partitions and model settings are detailed in the available matrix files. The Mk model [98] was used for the morphological data partition within our combined data matrix. Bayesian analyses [99] of molecular and combined data were performed in MrBayes (Version 3.2.7a [100,101]) via the CIPRES portal [102]. MrBayes settings used were default, with the exception of running the analysis for 40,500,000 generations. The Bayesian analysis code as well as all matrices and output files are included in the Supplementary Data (via 10.5061/dryad.v15dv4208).

## Results

Morphological results denote the results that were acquired from parsimony analysis of the morphological data, whereas the combined results comprise results of Bayesian analysis of the combined morphological and DNA data. *Paakniwatavis grandei* Musser and Clarke sp. nov. urn:lsid:zoobank.org:act:05722EDF-EDF1-4B2C-87A0-493D97DC7B3D is recovered across all analyses as the sister taxon of Anseranatidae+Anatidae (~50% bootstrap value in both morphological results, 85% clade credibility in both combined results (Figs 5, S1 and S2 in Supplementary Data via 10.5061/dryad.v15dv4208).

Across both the parsimony analysis of morphological data and the Bayesian analysis of combined data, addition of *Wilaru* results in differing topologies and causes nonmonophyly of Galloanseres. Due to this, the monophyly of Galloanseres was minimally constrained using a molecular backbone constraint when *Wilaru* was included in morphological analysis, and the following results discussed with *Wilaru* included comprise a monophyletic Galloanseres (S1 and S2 Figs, Supplementary Data via 10.5061/dryad.v15dv4208).. In the morphological results, *Wilaru* is placed as the sister taxon of *Gastornis* within a stem anseriform clade that also contains the Pelagornithidae (<50% bootstrap support; S2 Fig, Supplementary Data via 10.5061/dryad.v15dv4208). Addition of *Wilaru* also changes the relationships within the clade containing *Vegavis iaai*, *Telmabates*, *Presbyornis*, *Conflicto*, and *Nettapterornis 32xford* that is sister to Anatidae recovered across all analyses. With *Wilaru* excluded, *Telmabates* and *Presbyornis* are placed as sister taxa and form the sister clade to a group comprising *Vegavis*(*Conflicto+-Nettapterornis*), whereas inclusion of *Wilaru* results in *Vegavis* being placed as the sister taxon of a clade containing *Telmabates*(*Presbyornis*(*Conflicto+Nettapterornis*)). When *Wilaru* is included, bootstrap support scores rise to 99% per node within the group *Presbyornis*(*Nettapterornis+Conflicto*). Additional topology changes when *Wilaru* is included in the

morphological analyses include *Dendrocygna guttata* being placed as the sister taxon of *Thalassornis leuconotus* and differences in the placements of several additional crown anatids (*Mergus serrator*, *Netta rufina*, *Cygnus atratus* and *Coscoroba coscoroba*; S1 and S2 Figs, Supplementary Data via 10.5061/dryad.v15dv4208).

When a Galloanserine molecular backbone constraint was not enforced, *Danielsavis nazensis* was recovered as the sister taxon of *Anachronornis anhimops* (77% bootstrap support) and placed within a group comprising Pelagornithidae(*Gastornis*(*Anachronornis*+*Danielsavis*)) (<50% bootstrap support; S4 Fig, Supplementary Data via 10.5061/dryad.v15dv4208). This clade was placed within a polytomy containing *Ichthyornis dispar* and the rest of included Aves. *Perplexicervix microcephalon* was recovered as the sister taxon to Pan-Anseriformes (<50% bootstrap support). When *Perplexicervix* was excluded, *Danielsavis* was again placed as the sister taxon of *Anachronornis* (83% bootstrap support); however, the clade including these taxa, the Pelagornithidae and *Gastornis* collapsed into a polytomy containing these taxa (S5 Fig, Supplementary Data via 10.5061/dryad.v15dv4208). When Galloanserine molecular backbone constraints were enforced, *Danielsavis*, *Anachronornis* and *Perplexicervix* were recovered in a polytomy with *Ichthyornis* and the rest of included Aves (S6 Fig, Supplementary Data via 10.5061/dryad.v15dv4208). Exclusion of *Perplexicervix* did not change these results except that *Anachronornis* and *Danielsavis* were recovered as sister taxa (77% bootstrap support; S7 Fig, Supplementary Data via 10.5061/dryad.v15dv4208).

In the combined results, addition of *Wilaru* further defines the base of included Galliformes recovered in the morphological analyses (*Gallinuloides wyomingensis*(*Macrocephalon maleo* +*Megapodius freycinet*)) and collapses the base of a clade containing *Telmabates*, *Presbyornis*, *Nettapterornis*, *Conflicto*, *Vegavis* and *Wilaru* that is sister to Anatidae into a polytomy (94% clade credibility; S3 Fig, Supplementary Data via 10.5061/dryad.v15dv4208. Both combined results recovered a *Presbyornis*(*Nettapterornis*+*Conflicto*) clade, with posterior probability values for this clade rising when *Wilaru* is included. When *Wilaru* is excluded, *Vegavis* is placed as the sister taxon to a *Telmabates*(*Presbyornis*(*Nettapterornis*+*Conflicto*)) group, as in the morphological results with *Wilaru* included, although the clade credibility of this recovered group is low (85% or less, see Figs 5 and S3, Supplementary Data via 10.5061/dryad.v15dv4208). These drastically different placements across results from both the combined and morphological datasets are likely due to the lack of a skull known for *Wilaru*, although it does suggest that *Wilaru* may not be a presbyornithid as previously hypothesized in De Pietri et al. [58]. Due to this, we will largely focus on discussing analyses that exclude *Wilaru*.

Results from Bayesian analysis of the combined data matrix with *Wilaru* excluded are presented in Fig 5. Placement of extant taxa in the combined results are consistent with the resulting phylogram of Sun et al. [31] based on mitochondrial genomes; however, this tree included relatively few anseriform taxa used in our analysis. Combined results remain largely consistent with those of the phylogram of Sun et al. [31] based on two mitochondrial genes, which contains almost all extant anseriform taxa used in our analysis. Differences arise in the placement of several non-goose taxa within crown Anatidae. We recover a clade containing *Oxyura*+*Stictonetta* (*Anas*+*Amazonetta*(*Netta*(*Mergus*(*Tadorna*+*Chloephaga*)))) as being sister to the goose clade *Cygnus*+*Coscoroba*(*Branta*(*Anser*+*Anser*)), whereas Sun et al. [31] recovers a *Mergus* (*Tadorna*+*Chloephaga*(*Netta*(*Anas*+*Amazonetta*))) group that is sister to *Oxyura*+the goose clade. Differences between these analyses are likely due to our inclusion of morphological data, mitochondrial genomes for more taxa and nuclear genes. Additionally, the Sun et al. [31] analysis did not include *Stictonetta*, *Coscoroba*, or *Thalassornis*, and our analysis did not include several genera included in Sun et al. [31] (*eg. Neochen*, *Melanitta*, *Aythya* and more).

Within the combined results (Fig 5), clade credibility values of all nodes were 94% or higher with several exceptions. Exceptions comprise placement of *Asteriornis maastrichtensis* as the

sister taxon of a Tinamidae+Lithornithidae clade (54%), placement of *Crax* as the sister taxon of a *Lophura*+*Gallus gallus* sister group (69%), placement of *Paakniwatavis grandei* as the sister taxon to *Anseranas semipalmata*+(a fossil clade+crown Anatidae) (85%), placement of *Anseranas* as the sister taxon to a a clade that contains extinct taxa and crown Anatidae (89%), placement of *Telmabates* as the sister taxon to *Prebyornis*(*Conflicto*+*Nettapterornis*) (59%), placement of *Presbyornis* (55%), placement of *Nettapterornis* as the sister taxon of *Conflicto* (88%), and placement of *Netta rufina* as the sister taxon to a clade containing *Mergus*(*Tadorna* +*Chloephaga*) (85%).

The strict consensus tree from analysis of morphological data is broadly congruent with combined results but presents alternative hypotheses for the placement of several extinct and extant taxa within Pan-Anatidae, the name we are giving to the clade that includes crown Anatidae and the clade of extinct taxa that was placed as the sister taxon to crown Anatidae across all analyses (S1 Fig, Supplementary Data via 10.5061/dryad.v15dv4208). Unconstrained parsimony analysis of morphological data resulted in one most parsimonious tree (MPT) of 2,821 steps (CI = 0.291, RI = 0.580, RC = 0.169, HI = 0.709). Morphological analysis recovers a clade containing *Telmabates*+*Presbyornis*(*Vegavis*(*Conflicto*+*Nettapterornis*)) as the sister taxon to Anatidae. This clade containing the same taxa is structured as follows within the combined data results: *Vegavis*(*Telmabates*(*Presbyornis*(*Conflicto*+*Nettapterornis*))). The positions of several extant anatids differ across the analyses as well. In the morphological results, *Dendrocygna* is sister to *Thalassiornis*+crown Anatidae, whereas *Dendrocygna* and *Thalassiornis* are sister taxa in the combined data results. Additional differences within the morphological results comprise the placements of *Netta*, *Oxyura*, *Stictonetta*, the *Chloephaga*+*Tadorna* sister group, and *Mergus*. *Cygnus* and *Coscoroba* are additionally no longer sister taxa in the morphological results.

Analysis of the combined dataset and morphology only dataset resultin *Gastornis* (*Pelagornis*+*Protodontopteryx*) as a stem anseriform clade (<50% bootstrap support, 99% clade credibility;(Figs 5 and S3, Supplementary Data via 10.5061/dryad.v15dv4208). Both datasets also recover a clade containing *Telmabates*, *Presbyornis*, *Vegavis*, *Conflicto*, and *Nettapterornis* (<50% bootstrap support, 95% clade credibility) as the sister taxon of Anatidae. Recovery of this clade containing solely extinct taxa may be real or discovered to be a paraphyletic assemblage; all synapomorphies for this clade exhibited CI≤0.5 in the morphological results and CI≤0.5 in the combined data results (see Supplementary Data via 10.5061/dryad.v15dv4208).

Ancestral state reconstruction using the combined dataset indicates that *P. grandei* likely preferred an aquatic or semi-aquatic environment, was either not specialized for aquatic feeding modes (hereafter referred to as a "non-swimmer") or was a surface swimmer, was primarily herbivorous, was a grazer, had some form of rhamphothecal lamellae, had a pedal digit III that was subequal to or longer than that of the tarsometatarsus, had an ossified pessulus of the syrinx, and did not have asymmetry at the tracheobronchial juncture of the syrinx. Status of pedal webbing could not be reconstructed for *P. grandei*; however, the anatomy and skeletal proportions of *P. grandei* suggest that it was an aquatic surface swimmer. This is especially likely as *P. grandei* has a pedal digit III that is longer than the tarsometatarsus, indicating that this taxon likely was a surface swimmer and led an aquatic or semi-aquatic lifestyle [103]. Optimizations for the Anseriformes total group including stem-Anseriformes indicate that this clade preferred either a terrestrial or aquatic habitat, were non-swimmers, were omnivorous or carnivorous, were mixed feeders or grazers, did not have pedal webbing, had a pedal digit III was shorter than the length of the tarsometatarsus, had an ossified pessulus, and had no asymmetry at the tracheobronchial juncture. The status of rhamphothecal lamellae could not be reconstructed for this clade. Optimizations for crown Anseriformes were identical to those for total group Anseriformes with the exception of crown Anseriformes preferring an aquatic

or semiaquatic habitat, being primarily herbivorous, having a form of rhamphothecal lamellae, and being grazers. Optimizations for Anatidae and Pan-Anatidae indicate that they originally preferred an aquatic or semiaquatic habitat, were surface swimmers, were primarily herbivorous, had full rhamphothecal lamellae present, were grazers, possessed pedal webbing, had a pedal digit III was subequal in length or longer than the tarsometatarsus, had an ossified pessulus, and had no asymmetry at the tracheobronchial juncture. Optimizations for crown anatids were identical with the exception of optimizations suggesting that they preferred aquatic environments only.

Ancestral state reconstruction using the morphological data provided identical results for *P. grandei*. Status of pedal webbing again could not be reconstructed for *P. grandei*. Optimizations for total group Anseriformes were identical to those of the combined data analysis. Again, the status of rhamphothecal lamellae could not be reconstructed for this clade. Optimizations for crown Anseriformes remained identical. Optimizations for Anatidae and Pan-Anatidae remained identical with the exception of asymmetry at the tracheobronchial junction being unable to be reconstructed. Optimizations for crown Anatidae were identical to those recovered using the combined data results, with the exception of asymmetry at the tracheobronchial junction being unable to be reconstructed.

## Discussion

We recover *Paakniwatavis grandei* as the sister taxon to Anseranatidae across all analyses, regardless of taxon sampling. This placement is consistent with the unique combination of anhimid-like and anseranatid-like morphologies displayed by the taxon as well as its aquatic morphologies. A *Gastornis*+Pelagornithidae group was also recovered across all analyses as the sister taxon of crown Anseriformes, and a clade containing *Vegavis iaai*, *Presbyornis*, *Telmabates*, *Conflicto*, and *Nettapterornis* was recovered as the sister taxon of crown Anatidae across all analyses (called Pan-Anatidae here). *Asteriornis maastrichtensis* Field et al. 2020 is recovered as a stem Galloanserine in the morphology only topology (<50% bootstrap support), but is the sister taxon to a Tinamidae+Lithornithidae clade within the combined results (54% clade credibility).

Our results for *Gastornis* and the Pelagornithidae are consistent with those of Bourdon [24], in which Pelagornithidae are the sister taxon of Anseriformes; however, this study did not recover a monophyletic Galloanseres. Field et al. [11] recovered the Pelagornithidae either as the sister taxon to an Anseriformes(*Conflicto*+*Nettapterornis*) group or within a polytomy containing Neoaves and Galloanseres. Mayr [25] recovered Pelagornithidae as the sister taxon to a Sylviornithidae(Dromornithidae(Galloanseres))) group, and Mayr et al. [51] recovered a polytomy comprising Pelagornithidae, Galloanseres and Neoaves. Using the dataset of Field et al. [11], Houde et al. [44] recovered Pelagornithidae as the sister taxon to a small sample of four neoavian species using Bayesian analysis; however, when parsimony analysis was used on the same dataset Galloanseres is nonmonophyletic (see Supplement of [44]), so much so that Galliformes are nested within Palaeognathae and the Pelagornithidae are sister to a polytomy containing the four included neoavian species and Anseriformes (including Pan-Anatidae). This level of non-monophyly calls the legitimacy of the Field et al. [11] dataset into question and indicates that character revision and/or revision of the scorings of this dataset are necessary before use in future studies.

Placement of *Presbyornis*, *Nettapterornis*, and *Vegavis* within a clade that is sister to Anatidae is consistent with the results of Ericson [14], Livezey [13], and Elzanowski and Stidham [52] for *Presbyornis* and both *Vegavis* and *Presbyornis* in the Livezey [13] matrix (see also Clarke et al. [23]). Worthy et al. [27] recovered *Presbyornis* and *Wilaru* as either the sister group of *Anseranas*+Anatidae or *Anseranas*, and recover *Vegavis* as either the sister taxon of

the anseriform-like Gastornithiformes (eg. *Gastornis*) or the sister taxon to Anseriformes. Tambussi et al. [28], using the Worthy et al. [27] data matrix, similarly recover *Presbyornis*, *Wilaru*, *Nettapterornis* and *Conflicto* as stem Anseriformes and place *Vegavis* as the sister taxon of Gastornithiformes within stem Anseriformes. Field et al. [11] recover *Wilaru*, *Presbyornis*, *Conflicto*, and *Nettapterornis* as stem Anseriformes outside Anhimidae+Anatidae or *Wilaru* and *Presbyornis* as the sister group of *Anseranas*. Field et al [11] also recovered *Vegavis* as a stem Galloanserine or Neoavian taxon (it remains in an unresolved polytomy) or as a stem Neornithine, and place *Asteriornis* as a stem Galloanserine or stem galliform. Torres et al. [104] places *Conflicto* and *Vegavis* within a polytomy containing Anatidae and recovers *Asteriornis* as the sister taxon of a *Lithornis*+tinamou clade.

*Anachronornis anhimops* and *Danielsavis nazensis* were placed as sister taxa across almost all morphology only analyses (S4, S5 and S7 Figs, Supplementary Data via 10.5061/dryad.v15dv4208), and were placed within a clade containing *Gastornis gigantea* and the Pelagornithidae in results where *Wilaru* was excluded and Galloanserine molecular backbone constraints not enforced (S4 and S5 Figs, Supplementary Data via 10.5061/dryad.v15dv4208). *Perplexicervix microcephalon* was recovered as the sister taxon to Pan-Anseriformes in results where *Wilaru* was excluded and Galloanserine molecular backbone constraints not enforced (S4 Fig, Supplementary Data via 10.5061/dryad.v15dv4208). Houde et al [44] recover *Anachronornis* and *Danielsavis* as stem Anseriformes using both parsimony analysis and Bayesian analysis with low support. When excluding *Danielsavis* from the employed datasets, Houde et al. [44] place *Anachronornis* as either a stem anseriform, stem anhimid, stem anseranatid, or within a polytomy containing other Anseriformes and/or stem Anseriformes depending on the dataset and analysis method used. Thus the phylogenetic placement of both extinct taxa remains tenuous at best and will require further study.

*Perplexicervix* has not been included in any previously published phylogenetic analysis. *Perplexicervix* was originally thought to be anseriform [45]; however, Mayr et al. [60] remarked on a few similarities between *Perplexicervix* and Otididae but did not perform phylogenetic analysis that included this taxon or identify unambiguous character evidence to support this hypothesis. Like Galloanserines, *Perplexicervix* exhibits basipterygoid processes, which are absent in most of Neoaves. Combined data results yielded 7 unambiguous and 3 ambiguous optimized synapomorphies of the quadrate, coracoid, scapula, humerus, femur, tibiotarsus, tarsometatarsus, and pedal phalanges (two with CI = 1) that support placement of *P. grandei* within Anseriformes. Within the quadrate, the ventral apex of the crista tympanica terminates within the ventral half of the quadrate (character 172: state 2, ambiguous). The labrum externa along the lateral angle of the sternal coracoid is ventrocranially angled (416:2, unambiguous). Within the scapula, the acromion process is truncate and does not reach cranially beyond the articular faces of the head (437:1, ambiguous). The tuberculum ventral and the crista along the proximal margin of the fossa pneumotricipitalis are domed and distally prominent, overhanging the fossa pneumotricipitalis (456:2, unambiguous, CI = 1). The length of the femur is approximately one half the length of the tibiotarsus (597:1, unambiguous). The distal opening of the pons supratendineus of the tibiotarsus is centered along the midline (653:3, unambiguous). Within the tarsometatarsus the medial margin of the medial cotyle is exceptionally projected proximally and crista-like (658:2, unambiguous, CI = 1), the lateral cotyle is flattened or only slightly concave (662:2, ambiguous), and the hypotarsal eminence is proximally prominent (664:2, unambiguous). Within the pedal phalanges, pedal phalanx II: digit 2 is slightly more elongate than III:2 (706:2, unambiguous).

Seven unambiguous and three ambiguous optimized synapomorphies of the quadrate, mandible, furcula, scapula, carpometacarpus, tarsometatarsus and pedal phalanges (one with CI = 1) support placement of *P. grandei* as a sister taxon to the clade (Anseranatidae

+Anatidae). Within the quadrate, the crista tympanica is present and exhibits an extremely prominent crista (171:3, unambiguous), and the caudal face of the otic process is deeply concave (176:2, unambiguous). Within the mandible, the medial portion of the ramus caudal to the coronoid process (or homologous site) is extremely deep and concave (244:2, unambiguous), and a true retroarticular process is present and exceptionally tapered throughout (256:2, ambiguous). The width of the lateral diameter of the furcular ramus is larger than that at the symphysis (436:3, ambiguous), and the scapula is shorter than the humerus in length (449:1, unambiguous). Within the carpometacarpus, the proximal terminus of the dorsal rim of the trochlea carpalis is strongly angular and proximally elongated (514:2, unambiguous). Within the tarsometatarsus, the distal-most terminus(i) of medial crest(s) are much more distally extensive and the lateral crista(e) are proximodistally truncate, about 1/2-2/3 proximodistal length of the medial crista (669:1, unambiguous, CI = 1). The proximal portion of the sulcus extensorius medial to the dorsal foramina vascularia proximalia is present and deeply excavated (680:2, ambiguous). Within the pedal phalanges, phalanx III is longer than the tarsometatarsus (719:1, unambiguous).

Our phylogenetic placements of the abovementioned extinct taxa suggests that extinct anseriform diversity is likely much more vast than previously known and remains poorly understood. Likely plesiomorphic traits shared by these taxa are additionally necessary to further explore and understand to better resolve the relationships and evolutionary histories of these taxa, especially as new relevant fossils are discovered. Analysis of the Cretaceous ornithurine *Janavis finalidens* [105] and recent evidence on the ichthyornithine palatine [104] similarly indicate that reevaluation of purported galloanseran affinities of early Cenozoic groups is necessary, especially within the Pelagornithidae. Recovery of additional, more complete and better preserved anseriform-like fossils is ultimately necessary to more robustly resolve the phylogenetic placement of these important taxa. Adding Neoaves to the dataset is also important in future iterations of the Musser and Cracraft [79] dataset. Despite this, consistent results across several analyses using different data types and methods suggest that most placements of the included taxa are fairly robust. *P. grandei* represents a unique ecology for the Green River Formation, and this new fossil along with this new dataset and re-evaluation of *Presbyornis* material thus begins to elucidate several critical issues in anseriform evolution.

Ancestral state reconstruction across all analyses suggests the evolution of a combination of terrestrial and semi-aquatic traits at the base of Anseriformes, with crown Anseriformes exhibiting a shift toward more aquatic traits such as preferring an aquatic or semiaquatic habitat, being primarily herbivorous, having a form of rhamphothecal lamellae, and being grazers (Fig 5). This is inconsistent with the assertion that Anseriformes were ancestrally terrestrial as suggested by Olson and Feduccia [22], Ericson [14], and Livezey [13]. The placement of *P. grandei* and its influence on the optimization of these reconstructions suggests aquatic or semi-aquatic ancestry of Anseriformes, especially combined with a reduced form of rhamphothecal lamellae present in extant Anhimidae [22].

The question of how anseriform beak morphology and filter feeding evolved remains an open one. Either a narrower, "goose-like" beak was ancestral for Anseriformes, or this narrower beak evolved several times within Anseriformes [19]. The "goose-like" beak is associated with increased leaf consumption, decreased invertebrate consumption, and an increase in the mechanical advantage of the beak that allows for more effective cropping of plants [19], whereas "duck-like" beaks are associated with increased filter feeding and consumption of invertebrates. We take this classification of a "goose-like" beak a step further in the broader context of both stem and crown Anseriformes: We first identify whether the rostrum is mediolaterally wider than the width of the paroccipital processes (indicating an anseranatid-like beak; character 3) and, if so, whether it is anteriorly tapered and narrowed further (a "goose-

like" beak; character 4: state 1), or whether it remains subequal in width (a "duck-like" beak; 4:2). *P. grandei* is the only Paleogene anseriform currently known to present a narrow, anhimid-like beak other than the Pelagornithidae, *Danielsavis* and *Anachronornis*. Its beak is mediolaterally narrower than the width of its paroccipital processes, as in extant Anhimidae. *Chaunoides* and *Telmabates* have no known preserved skull and *Vegavis* has no known preserved rostrum or braincase, while other Paleogene fossils with a beak preserved such as *Presbyornis*, *Nettapterornis* and *Conflicto* all present an anseranatid-like beak based on currently known remains. All of our analyses posit the first appearance of a beak that is mediolaterally wider than the width of the paroccipital processes as ancestral to the node containing *Anseranas semipalmata*. It would have been an anteriorly tapered, "goose-like" beak like that of the extant Anhmidae or *Anseranas* (Fig 5). Across all analyses, ancestral state reconstruction indicates that this "goose-like" beak is ancestral to Anseranatidae+Anatidae. All analyses indicate that anteriorly wider "duck-like" beaks evolve at least twice: once after the divergence of *Anseranas* in taxa closer to Anatidae (present in *Presbyornis* and *Nettapterornis*), and once within crown Anatidae (see Fig 5 and Supplementary Data via 10.5061/dryad.v15dv4208). These results contradict those of Olsen [19], who found that a duck-like beak was ancestral for most Anatidae, followed by multiple transitions toward a goose-like beak; however, this study performed ancestral state reconstruction using a phylomorphospace of beak curvature measurements, beak function metrics, and quantified diet data for a smaller sample of anseriform taxa that included only two extinct taxa, *Presbyornis* and the recently extinct moa-nalo *Thambetochen chauliodous* Olson and Wetmore [106] (see also Olson and James [107]). More robust placement of *Anachronornis* and *Danielsavis* in future studies will be critical to better understanding anseriform ancestral state reconstructions and the evolution of beak morphology.

At the same time, our results are consistent with the results of Olsen [19] in that we find rhamphothecal lamellae to have been present ancestrally for crown Anseriformes and Anatidae (including within both the stem and crown lineages of these groups), indicating that herbivory and/or filter feeding was ancestral for these clades. This again contradicts the hypothesis that Anseriformes were ancestrally terrestrial and would explain the presence of reduced rhamphothecal lamellae in extant Anhimidae [22]. Anhimidae represent the only known example of rhamphothecal lamellae being present without pedal webbing in extant birds; however, similar lamellae-like ridges have been found in Ornithomimus [108,109], Gallimimus, chelonians, hadrosaurs [109] and an edentulous ceratosaur [110]. A partial correlation between the presence of these ridges and exclusively herbivorous diet among terrestrial chelonians has been found [111,112], suggesting that more prominent ridges were present when more coarse vegetation was eaten. Studies on the jaw mechanics, locomotion and gut contents of hadrosaurs have similarly demonstrated that they were obligate terrestrial herbivores that used their beak for cropping through vegetation [113,114]. If our ancestral state reconstructions for crown Anseriformes are correct, lamellae coevolved with a shift toward herbivory and grazing along with the preference for a more aquatic habitat despite a lack of pedal webbing. Based on the available evidence and our results, some form of rhamphothecal lamellae was ancestral to crown Anseriformes and could have developed due to aquatic grazing, then remained (or became reduced) within extant Anhimidae while developing further within more derived crown anseriform taxa. Webbing then was maximally ancestral to *P. grandei*, and minimally was ancestral to *Anseranas*. Our results thus suggest that crown Anseriformes were maximally ancestrally aquatic or semi-aquatic, and fully aquatic ancestral to crown Anseranatidae. In general our results suggest a trend within Anseriformes toward aquatic grazing and the anseranatid "goose-like" beak to obtain an herbivorous (increased leaf and root consumption) diet, with at least two evolutions of a "duck-like" beak associated with increased filter feeding and invertebrates obtained by this feeding mode at least once within

Pan-Anatidae and once within Anatidae [115,116]. If these placements and ancestral state reconstructions are correct, this would add to mounting support that feeding ecology has acted as the primary selective force in waterfowl beak shape diversification [19].

Further elucidation of anseriform behavioral evolution is indicated in ancestral state reconstruction of syringeal characters. Ancestral state reconstruction within the combined results indicates that *P. grandei*, Anseriformes (inclusive and exclusive of stem Anseriformes) and anatids (inclusive and exclusive of the clade containing extinct taxa that was placed as the sister to Anatidae) had an ossified pessulus of the syrinx, a derived neognath bird feature that has been proposed to anchor enlarged vocal folds or labia [117], consistent with the results of Clarke et al. [42]. These taxa also were indicated to ancestrally not possess asymmetry at the tracheobronchial juncture of the syrinx. Ancestral reconstruction within the morphological results is identical with the exception of ambiguity within Pan- and crown Anatidae regarding asymmetry at the tracheobronchial juncture. Both results indicate that asymmetry at the tracheobronchial juncture must have evolved at least once within Anseranatidae+Anatidae. This is somewhat consistent with Clarke et al. [42]; however, while Clarke et al. [42] considered a single origin in Anatidae likely, our results may suggest more gains and losses within this Anseranatidae+Anatidae clade. Further study and coding of extant syrinx asymmetry is necessary as previous descriptions and figures of extant syrinxes largely focus on pronounced asymmetrical bullae in some male anseriform taxa rather than the more subtle asymmetry of the rings found in females, and the large range of variation across differing taxa and sexes within Anseriformes is not well understood [20,117]. Better understanding asymmetry in extant taxa has important implications for extinct taxa as well; for example, the subtle asymmetry present in *Vegavis* may suggest that this taxon exhibited sexual dimorphism within the syrinx, as in some extant Anatidae. Asymmetry is an important trait to further study as it is correlated with the presence of a dual sound source and the presence of labia [42,117]. Recovery of further fossils that include syrinxes and further study of extant syrinx anatomy and function in extant birds is thus necessary to understand the evolution of this organ.

All analyses recover the K-Pg taxa *Vegavis*, *Nettapterornis*, *Conflicto*, *Presbyornis* and *Telmabates* within a clade that is the sister taxon to Anatidae. Within the combined data results, six unambiguous and 10 ambiguous synapomorphies (CI < 1) were recovered for this clade that united three or more of these taxa within the axial skeleton and hindlimbs (see Supplementary Data via 10.5061/dryad.v15dv4208). Although several appear to be plesiomorphic, these characters may represent evolutionary and biological/ecological significance pending recovery of key elements from taxa such as *Vegavis*, *Telmabates* and *Conflicto*.

*P. grandei* exhibits non-heterocoelous vertebrae. Although this character was not optimized as a synapomorphy for this clade, *Nettapterornis*, *Telmabates*, *Presbyornis*, and the Neogene Dromornithidae have amphicoelous thoracic vertebrae [14,56,118], whereas *Vegavis*, Pelagornithidae and gastornithids have heterocoelous thoracic vertebrae [23,42]. Although it cannot be discerned which non-heterocoelous form the vertebrae of *P. grandei* possess due to taphonomic distortion, it is likely that *P. grandei* possessed amphicoelous vertebrae as well. Amphicoelous thoracic vertebrae are plesiomorphic within Avialae but also present in well-nested neoavian clade Charadriiformes [14,56,88,118]. Opisthocoelous thoracic vertebrae are only known within Neoaves [79,88,89], and amphiplatyan and procoelous vertebrae are only known in non-avian dinosaurs [89].

Further study on the function of amphicoelous vertebrae in the context of avian evolution and ecology is needed, especially since birds possess a unique dorsal intervertebral joint [119]; however, the literature on this in fishes and crocodylomorphs suggests that amphicoelous vertebrae provide a more rigid spine [120–122] that can withstand increased stress without deformation that may be caused by powerful movements of musculature [120,121], allowing for

rapid flexure of the spine. Amphicoelous vertebrae have evolved several times in sharks, dipnoans, bony ganoids and teleosts, associating their appearance with improved speed of motion [120]. Amphicoelous vertebrae are also associated with aquatic environments, as transitions from amphicoelous to platycoelous vertebrae has been hypothesized to represent a transition from aquatic to terrestrial environments [123].

Results further clarify the complex picture of avian evolution around the K-T boundary, indicating that several lineages within Anseriformes with a variety of ecologies not represented in the crown were present by the latest Cretaceous and into the early Paleogene. *P. grandei* represents an early Eocene lacustrine taxon that was likely aquatic, swam and used its narrow bill in aquatic grazing or mixed feeding. The Cretaceous-early Paleogene *Nettapterornis* and *Presbyornis* were aquatic taxa that likely filter fed on a more invertebrate-heavy diet within both marine and marine and lacustrine environments, respectively [19]. The early Eocene *Telmabates* may have shared a similar ecology to *Presbyornis* given its amphicoelous thoracic vertebrae and evidence that the Casamayor formation in which it was found is known to be a marine-fluvial transition zone [124]. Other Cretaceous and Paleogene material has been referred to *Presbyornis* from more fluvial as well as marine settings possibly suggesting a cosmopolitan, flexible habitus [125–128]. At the same time the Cretaceous *Vegavis*, with heterocoelous thoracic vertebrae, was present in a near shore marine environment with unknown diet, and the Cretaceous *Conflicto* was present within a near shore marine/transitory estuarine environment, again with unknown diet and locomotion due to missing data [28]. In addition to this array of taxa and ecologies, the specialized Cretaceous-Paleogene marine, piscivorous pseudodontorns and the giant terrestrial Paleogene gastornithids were present within the stem anseriform lineage. If *Vegavis* and *Conflicto* are also found to have had omnivorous, mixed and/or piscivorous diets, a proliferation of Cretaceous-early Paleogene non-herbivorous stem and crown Anseriformes may have arisen. Proposed significant loss of plant cover due to global cooling and the K-T impact event [129–134] could suggest a short but strong selective regime favoring mixed and non-herbivore specialists. Fossil evidence suggests that early Anseriformes were diversifying rapidly since at least the late Cretaceous and were already widespread within the same time frame, as the early Eocene *Telmabates* was found in Patagonia and Paleocene and Eocene *Presbyornis* material has been recovered from North America, Europe and Mongolia [33,126–128].

An approximate mean body mass estimate for *P. grandei* is 304g based on the published allometric equation using femoral length from Field et al. [81]. This body mass level is quite small; it is most comparable to the mass of many *Anas* (within the 300s range). Many other anatids are generally larger (which typically range from 600-over 1,000g) [135]. Its body mass is estimated to be less than half that of *Presbyornis* and *Telmabates* (882g and 1423g, respectively). This is consistent with recent evidence that correlation between herbivory and body mass is not significant when accounting for phylogeny [18].

Further analysis of these Paleogene anseriform fossils in the context of broader extinct taxon sampling, especially in the context of extinct anseriform-like taxa, is necessary to further evidence placement of *P. grandei* and other Paleogene anseriform-like taxa and to gain more robust insight into ancestral states; however, *P. grandei* represents a key taxon, a unique ecology within known Anseriformes and the Green River Formation, and a potential calibration point for anseranatids. Other included extinct taxa also represent potential calibration points that would be valuable for stem Anseriformes, Anhimidae and Pan-Anatidae. Recovery of additional fossils and further phylogenetic analyses (especially of taxa such as *Chaunoides* and *Wilaru*) are preferable to confirm these relationships, further reveal the ecological and behavioral evolution and biogeography of Anseriformes, and better elucidate our understanding of avian evolution.

## Supporting information

**S1 Fig. Strict consensus trees recovered through parsimony analysis of morphological data.** Strict consensus tree of 1 MPT of 2,821 steps recovered based on morphological data with *Wilaru* excluded (CI = 0.291, RI = 0.580, RC = 0.169, HI = 0.709). Bootstrap support values greater than 50% are denoted above branches. Transitions are mapped for selected key nodes based on ancestral state reconstruction. Transitions are only mapped where ancestral state reconstruction is definitive.
(JPG)

**S2 Fig. Strict consensus tree of Strict consensus tree of 1 MPT of 2,928 steps recovered based on morphological data with *Wilaru* included (CI = 0.281, RI = 0.561, RC = 0.158, HI = 0.719).** Bootstrap support values greater than 50% are denoted above branches. Transitions are mapped for selected key nodes based on ancestral state reconstruction. Transitions are only mapped where ancestral state reconstruction is definitive.
(JPG)

**S3 Fig. Resulting consensus tree from Bayesian analysis of 719 morphological characters and 158,368 base pairs with *Wilaru* included (A) and excluded (B).** Clade credibility values greater than 50% are annotated above branches. Transitions are mapped for selected key nodes based on ancestral state reconstruction. Transitions are only mapped where ancestral state reconstruction is definitive. Extinct taxa are delimited with daggers.
(TIF)

**S4 Fig. Strict consensus tree of 3 MPTs of 2,898 steps recovered based on morphological data.** Galloanserine molecular backbone constraints were not used. *Wilaru* was excluded and *Anachronornis*, *Danielsavis* (including specimen NMS.Z.2021.40.2) and *Perplexicervix* were included (CI = 0.288, RI = 0.584, RC = 0.168, HI = 0.712).
(JPG)

**S5 Fig. Strict consensus tree of 4 MPTs of 2,897 steps recovered based on morphological data.** Galloanserine molecular backbone constraints were not used. *Wilaru* and *Perplexicervix* were excluded and *Anachronornis* and *Danielsavis* (including specimen NMS.Z.2021.40.2) were included (CI = 0.288, RI = 0.579, RC = 0.167, HI = 0.712).
(JPG)

**S6 Fig. Strict consensus tree of 27 MPTs of 2,989 steps recovered based on morphological data.** Galloanserine molecular backbone constraints were used. *Wilaru* was excluded and *Anachronornis*, *Danielsavis* (including specimen NMS.Z.2021.40.2) and *Perplexicervix* were included (CI = 0.279, RI = 0.565, RC = 0.158, HI = 0.721).
(JPG)

**S7 Fig. Strict consensus tree of 1 MPT of 2,986 steps recovered based on morphological data.** Galloanserine molecular backbone constraints were used. *Wilaru* and *Perplexicervix* were excluded and *Anachronornis* and *Danielsavis* (including specimen NMS.Z.2021.40.2) were included (CI = 0.280, RI = 0.561, RC = 0.157, HI = 0.720).
(JPG)

**S1 Appendix.**
(DOCX)

## Acknowledgments

We thank all of the staff of FMNH, especially Lance Grande, Jingmai O'Connor, William Simpson, Adrienne Stroup, Shannon Hackett, John Bates, and Ben Marks for specimen access and valuable discussion. We thank April Neander for scanning the specimen, and Matthew Colbert for aid in scan visualization and segmentation. We thank Helen James, Christopher Milensky, Brian Schmidt and Mark Florence of the Smithsonian National Museum of Natural History for access to the Ornithology and Vertebrate Paleontology collections. We thank Joel Cracraft, Paul Sweet, Mark Norell, Ruth O'Leary and Carl Mehling of AMNH for access to the Ornithology and Vertebrate Paleontology Collections. We additionally thank Kenneth Bader, Matthew Brown, and Christopher Sagebiel for access to additional TMM collections and their aid in working with AMNH loans. Finally, we thank Joel Cracraft, Zhiheng Li, Melissa Kemp, Christopher Bell, Daniel Field, Daniel Ksepka and Christopher Torres for valuable discussion.

## Author Contributions

**Conceptualization:** Grace Musser, Julia A. Clarke.

**Data curation:** Grace Musser.

**Formal analysis:** Grace Musser, Julia A. Clarke.

**Funding acquisition:** Grace Musser, Julia A. Clarke.

**Investigation:** Grace Musser, Julia A. Clarke.

**Methodology:** Grace Musser, Julia A. Clarke.

**Project administration:** Grace Musser, Julia A. Clarke.

**Resources:** Grace Musser, Julia A. Clarke.

**Software:** Grace Musser.

**Supervision:** Grace Musser, Julia A. Clarke.

**Validation:** Grace Musser, Julia A. Clarke.

**Visualization:** Grace Musser, Julia A. Clarke.

**Writing – original draft:** Grace Musser.

**Writing – review & editing:** Grace Musser, Julia A. Clarke.

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
