## [Decision Letter · Decision Letter 0]

8 Jan 2023

PONE-D-22-32093A new Paleogene fossil and a new dataset for waterfowl (Aves: Anseriformes) clarify phylogeny, ecological evolution, and avian evolution at the K-Pg Boundary.PLOS ONE

Dear Dr. Musser,

Thank you for submitting your manuscript to PLOS ONE. After careful consideration, we feel that it has merit but does not fully meet PLOS ONE’s publication criteria as it currently stands. Therefore, we invite you to submit a revised version of the manuscript that addresses the points raised during the review process. Two experts of Paleogene birds have accepted to comment on your manuscript. Both find the specimen very important, certainly to deserve publication in PLoS One but both are critical regarding its identification. They also have major concerns about the comparisons. The reviewers provided very detailed analysis of the manuscript, including numerous suggestions to help improve it.

I recommend paying particular attention to the following points of the reviews:

- Revaluate or reinforce the systematic attribution based on reviewers’ comments.

- Complete and extend the comparisons, especially with the Anhimidae.

- Make the synonymy of Diatryma with Gastornis (or explain why you disagree), which are not terror birds. This term characterizes another family of giant birds, the Phorusrhacidae.

- Several important papers dealing with the topic are missing. This is stated by the two reviewers. Please consider them in your revision.

- Please consider adding pictures of important anatomical details that are not visible in the present figures.

We look forward to receiving your revised manuscript.

Kind regards,

Thierry Smith, Ph.D.

Academic Editor

PLOS ONE

Journal Requirements:

2. Please take this opportunity to be sure you have met all of our guidelines for new species. For proper registration of a new zoological taxon, we require two specific statements to be included in your manuscript.

(1) In the Results section, the globally unique identifier (GUID), currently in the form of a Life Science Identifier (LSID), should be listed under the new species name, for example:

Anochetus boltoni Fisher sp. nov. urn:lsid:zoobank.org:act:B6C072CF-1CA6-40C7-8396-534E91EF7FBB

Another LSID for the manuscript itself should also appear within the Nomenclature statement. You will need to contact Zoobank (zoobank.org/About) to obtain a GUID (LSID). You should receive one LSID for your manuscript and a separate, unique LSID for the new species. 

(2) Please also insert the following text into the Methods section, in a sub-section to be called ""Nomenclatural Acts"":

The electronic edition of this article conforms to the requirements of the amended International Code of Zoological Nomenclature, and hence the new names contained herein are available under that Code from the electronic edition of this article. This published work and the nomenclatural acts it contains have been registered in ZooBank, the online registration system for the ICZN. The ZooBank LSIDs (Life Science Identifiers) can be resolved and the associated information viewed through any standard web browser by appending the LSID to the prefix ""http://zoobank.org/"". The LSID for this publication is: urn:lsid:zoobank.org:pub: XXXXXXX. The electronic edition of this work was published in a journal with an ISSN, and has been archived and is available from the following digital repositories: PubMed Central, LOCKSS [author to insert any additional repositories].

All PLOS ONE articles are deposited in PubMed Central and LOCKSS. If your institute, or those of your co-authors, has its own repository, we recommend that you also deposit the published online article there and include the name in your article.

Following a recent ruling by the International Commission on Zoological Nomenclature, electronic journals are now a valid format for publication of new zoological taxa. 

In order to ensure the valid publication of your new species, please be sure to include the updated version of Nomenclatural Acts (above). A complete explanation of our guidelines for publishing new species can be found on our website: http://www.plosone.org/static/guidelines#zoological.

"..We thank Joel Cracraft, Paul Sweet, Mark Norell, Ruth O’Leary and Carl Mehling of AMNH for access to the Ornithology and Vertebrate Paleontology Collections and thank the Ornithology Department of AMNH for the Ornithology Collections Study Grant that aided in funding this work.."

"This project was supported by a National Science Foundation GRFP (https://www.nsfgrfp.org/) award (to G.M., grant number DGE-16-4486), an Ornithology Collections Study Grant from the American Museum of Natural History (https://www.amnh.org/research/vertebrate-zoology/ornithology ;to G.M., 2019) and the Jackson School of Geosciences (https://www.jsg.utexas.edu/ ;G.M. and J.A.C). The funders had no role in study design, data collection and analysis, decision to publish, or preparation of the manuscript."

Reviewers' comments:

Reviewer's Responses to Questions

**Comments to the Author**

1. Is the manuscript technically sound, and do the data support the conclusions?

Reviewer #1: Partly

Reviewer #2: Partly

2. Has the statistical analysis been performed appropriately and rigorously? 

Reviewer #1: Yes

Reviewer #2: Yes

3. Have the authors made all data underlying the findings in their manuscript fully available?

Reviewer #1: Yes

Reviewer #2: No

4. Is the manuscript presented in an intelligible fashion and written in standard English?

Reviewer #1: Yes

Reviewer #2: Yes

5. Review Comments to the Author

Reviewer #1: You make extensive comparisons between the new taxon and extant and fossil duck-billed anseriforms (Presbyornis, Anatalavis, Anseranatidae, and Anatidae). However, just because of the absence of a duck-like beak, it is fairly obvious that the fossil is not very closely related to any of these taxa. What would instead be needed are more detailed comparisons with the Anhimidae, which is particularly true for the differential diagnosis, where Anhimidae are only mentioned once (and not differentiated from the new taxon). The description also mainly focuses on comparisons with non-anhimid taxa, which is difficult to understand given the fact that the fossil is much more similar to the Anhimidae than to the Anseranatidae and Anatidae.

Also, at the beginning of the results part you write that "Paakniwatavis grandei is recovered across all analyses as the sister taxon of Anseranatidae". However, this is not true, as is obvious from the tree you show, where it is the sister taxon of Anseranatidae + Anhimidae.

The omission, of the Anhimidae from most of your comparisons and discussion is difficult to understand and results in awkward statements like ” We recover Paakniwatavis grandei as the sister taxon to an Anseranas+Anatidae” (beginning of discussion). Actually, you recover the fossil as the sister taxon of an “Anhimidae + (Anseranatidae + Anatidae)) clade, and given the great similarity to screamers in some features (e.g., the massive furcula and the shape of the coracoid), I consider it highly likely that your fossil is a stem group anhimid.

I also do not quite understand, why any comparison with the Anhimidae are omitted from the discussion, as these certainly are the extant anseriform taxon that is most similar to the fossil with regard to skull morphology. You emphasize a potential significance of the new fossil for an understanding of anseriform evolution, but without any reference to the Anhimidae, this significance remains elusive. If this fossil is indeed a stem group anseriform, it would show that many features of the Anhimidae are plesiomorphic for anseriforms, as has been assumed by earlier authors. If it is a stem group anhimid, than the leg morphology may suggest that screamers were more aquatic in the past. In any case, a more balanced discussion is needed that includes more comparisons with the Anhimidae

What is needed is a more thorough discussion of the affinities pf the fossil. You note that “Seven unambiguous and three ambiguous optimized synapomorphies of the quadrate, mandible, furcula, scapula, carpometacarpus, tarsometatarsus and pedal phalanges support placement of P. grandei as a stem anseranatid.” However, what exactly does this mean? Are these characters present in crown group anseriforms but absent in the fossil? In the following text it rather reads as if these features were also present in the fossil, in which case they would not support a stem group position.

In my opinion, the comments on the evolution of syrinx morphology are out of place in the discussion, because the new taxon does not contribute to these issues. Given the unstable position of the fossil (and other anseriforms) in your analyses, ancestral state reconstructions are of rather limited value.

The fact that you recover a “Diatryma =Gastornis]+Pelagornithidae group” challenges the appropriateness your data set and it would be interesting to know what the character evidence for this grouping might be. Given the highly uncertain affinities of these fossils, I would also refrain from calling them “stem anseriform”, which is at least unlikely for pelagornithids.

I also believe that the title of the manuscript is not quite appropriate for this paper ("A new Paleogene fossil and a new dataset for waterfowl (Aves: Anseriformes) clarify phylogeny, ecological evolution, and avian evolution at the K-Pg Boundary"). Not only does your phylogenetic data set yield conflicting results, but I also do not see why this early Eocene fossil elucidates avian evolution at the K/Pg boundary. Actually, what you describe is a "screamer-like anseriform from the Green River Formation", in line with previous proposals that such birds were present in the early Eocene of North America (see comments above).

- throughout the manuscript: Why has the North American gastornithiform been assigned to the taxon Diatryma rather than to Gastornis? Most current authors synonymize both taxa.

p. 4, line 95: why do you consider the beak shape of the Paleocene Conflicto to be “largely unknown”? The holotype includes a nearly complete skull with nicely preserved beak.

- you should perhaps indicate that screamer-like Eocene anseriforms have already be identified by Peter Houde. These were mentioned by, e.g., Feduccia 1999 and Mayr 2009, 2017, 2022. It is likely that these fossils are closely related to your specimen. It would also be good to cite at least some of the aforementioned studies that indicate the presence of anhimid-like anseriforms in the early Eocene of North America.

- line 128: “partial skeleton and tracheal rings” – I would consider the tracheal rings to be part of the skeleton (usually one also do not separately mention, for example, scleral rings).

- line 130: you hypothesize that part of the skeleton has been” eroded”, by I think that this is an unfortunate wording. To me it rather appears as if the central part of the skeleton was dissolved by an acidic environment (possibly owing to the putrefaction of the trunk). Similar taphonomies are known from bog bodies and some birds from Messel.

- line 178: “dorsoventrally thick retroarticular process” – wouldn’t “dorsoventrally deep” be better (also in other places of the manuscript)?

- line 318: “cranio-medial aspect ” – this should be “cranio-ventral”

- line 320: The humeral head – what is meant here – the proximal end of the bone or the caput humeri? To avoid ambiguity, I suggest to use the proper anatomical terms throughout the manuscript.

- line 322: “tuberculum supracondylare ventrale” (add “e”)

- line 334: ” and has broken onto the skull.” – is this correct English? Sounds awkward to me.

- line 368: “supratendineus” (add “e”)

- line 371 “two sulci are present” – because you identify these sulci in the figure, some comments in the description would be appropriate. Also you may cite Mayr (201t5) for the identification of the hypotarsal sulci.

- line 373: “metatarsal trochlea III” – why not use the correct anatomical term (trochlea metatarsi III)?

- line 526: Presbyornis” (add “s”)

- line 566: “terror birds such as Gastornis” – gastornithiforms are no “terror birds”. This name is used for the Phorusrhacidae. It is also confusing that you use Gastornis here, whereas you classify the North American species as “Diatryma” (see above).

- line 586: “supratendineus” (substitute “o” with “e”)

- line 768 ff.: “all analyses recover the Cretaceous-Paleogene taxa Vegavis, Anatalavis, Conflicto, 769 Presbyornis and Telmabates within a clade that is the sister taxon to Anatidae”. Here, it would be good to note that some evidence against this classification exists, such as the absence of long retroarticular processes in Vegavis (see Mayr et al. 2018: On the taxonomic composition and phylogenetic affinities of the recently proposed clade Vegaviidae Agnolín et al., 2017 ‒ neornithine birds from the Upper Cretaceous of the Southern Hemisphere).

- line 784: “amphiplatyan“ – is this a correct word?

- line 1984: “Ericson, Per G. P.” – abbreviate the surname Per.

Reviewer #2: This paper represents an impressive study of a new and important fossil that bears on the early evolution of waterfowl. Overall, I have the impression that the authors have over-interpreted what is a preservationally compromised specimen. I am unable to fully assess whether the analyses have been undertaken rigorously as I do not have access to the author's data to independently verify it.

I hope my detailed comments below will prove helpful in necessary moderate revision.

23 - It is incorrect to say that Anseriformes have a comparatively rich, Cretaceous fossil

record

36-37 - Abstract – it is stated that the new Eocene taxon as a stem anseranatid across all analyses, yet it is not assigned or diagnosed as such. Instead it is simply Anseriformes

45 - Diatryma is a junior synonym of Gastornis

65 - I don’t recall that Olson and Feduccia 1980 claimed that Anhimidae possessed pedal webbing

89-90 - early Eocene Fossil Butte Member (FBM; 51.97 ± 0.16 Ma; Smith et al., 2010) of the Green River Formation

90-91 - The taxon was originally illustrated with a prospective referral by Storrs Olson to Heliornithidae (finfoots; Grande 2013); On the contrary, Grande included this fossil in his section “Waterfowl (Anseriformes…” He referred to this as an “undescribed possible waterfowl” and “an anseriform-like bird” (p225). There is no mention of Storrs Olson or heliornithids.

165 – “palaeognathid” is neither a family nor a word

167 – frigatebird is one word

168 – previously this is described as an unidentified ibis-like taxon

171 – the citation of Feduccia and Martin 1976 is odd since the birds they originally described have all been rediagnosed, without exception, yet the authors of this manuscript have avoided citing relevant original literature of other taxa. The literature review is selectively incomplete, and perhaps biased.

187-244 – numerous characters described in the Differential Diagnosis and Description are insufficiently illustrated for the reader to independently verify or even understand the features described. I cannot understand how details of the scapula or sternum can be inferred from Figure 1. I have not been given access to supplemental CT images. Without better documentation, I am left with the impression that the interpretations far outstrip the evidence.

191-192 - I do not believe that it can be shown that the bill was narrower than the bilateral width of the paroccipital processes on the basis of the crushed specimen.

198 – “blind pneumatic foramen” is an oxymoron

207 – Is there a pneumatic foramen to show that the fossa pneumotricipitalis of the humerus is pneumatic in Paakniwatavis?

213-214 – it is unclear what is meant by “The craniocaudal lengths of manual digits II and III at the synostosis are subequal in Paakniwatavis”

229 – medial mandibular process? retroarticular process?

233 - “is of”

233-234 - distal aperture of the sulcus extensorius, not the pons

240-241, 325, 548 – what is meant by “domed” crista

245 – Chaunoides is not italicized

248 – it is unclear what is meant by “the sternal facet curves cranially”

249 – the major metatarsal ridge does not appear to be hooked distally in Figure 4F1, F2

264 – In what way(s) are the crania of anhimids and Anseranas similar? This surprising statement needs elaboration.

265 – how can either jugal be evident if they are covered by the carpometacarpus and how could the left be evident if it is the right side of the skull that is exposed?

265-267 – it is not clear to me from the illustrations that the “lateroventrally projecting supraorbital crest is present”. Instead, it appears that the left prefrontal is broader than the supraorbital.

267 – that the postorbital process is (revealed by CT scans to be) elongate like that of most Anseriformes is particularly surprising since it is not in screamers, to which the bill of the new fossils resembles. The postorbital serves as an origin of jaw musculature that, among other characters, is associated with advanced filter feeding behavior of Anseriformes.

306-307 – stout relative to what directions?

309 – are there such things as ventral and dorsal clavicles?

313 – sternal (=proximal) or omal (=distal) extremities, not distal bases

314-315 - if CT images show the procoracoid to be present, then it should be included in Figure 4F

319 vs 230 – the descriptions of the deltopectoral crest seem a little contradictory

327 – ulnar body; than that of the humerus

329 – impressio brachialis

344 – check grammar

365 – is appears

377-383 – it may be true that femur length is the best (or only) correlate of body mass among those that that could be measured, but the scaling functions are sensitive to taxonomic group. The proportions of this fossil (e.g., pre vs postacetabular pelvis) are more like that of screamers than typical anatoids, on which the slope for Anseriformes is mostly if not entirely based. In my experience, the femora of even fossils preserved in the round produce mass estimates with error ranges as high as 100%. Thus, to report this estimate to the first decimal with no error range is unacceptable.

394 comprised of or include

467-504 – It is very interesting that the inclusion of Wilaru resulted in wide ranging discombobulation of the phylogeny, especially since it is so strongly supported as a member of Presbyornithidae! This, and the different mitochondrial results among anatoids that are a tad tangential to the subject here, might well be worthy of elaboration in a separate paper. What would be relevant to the subject of this manuscript, but is not reported, is whether similar eyebrow-raising results of the phylogenies reported here would accrue if Paakniwatavis was excluded. To what extent are these the effect of the fossil or of the character set? At any rate, these results cannot be dismissed as the result of Wilaru being know from only “highly fragmentary” remains, as its two species are in fact known from an abundance (embarrassingly large compared with, say, Vegavis, about which so much is written) of well preserved specimens.

530 congruent, not confluent

559-578 – in light of the different results reviewed in this paragraph, it would be a good idea to define what is meant by Anseriformes as it is used on line 578.

576, 592 – ambiguously optimized

594-595 - the crista

595 awkward sentence

613-614 – I am confused whether the length of the hallux was measured or inferred by ancestral state reconstruction. Regardless, if it was unusually long, then this would be similar to the condition in screamers and jacanas, which are noted for foraging from on top of floating vegetation.

625 – identical to what?

459, 643 – the authors used only 7 of the ~33 used by Hinić Frlog [misspelled throughout] and Montani in their study of skeletal correlates of swimming modes in their fossil. That Chauna was ambiguously recovered as a wing-propelled diver and that Degrange found similar morphospace among such ecologically dissimilar birds as cursorial/terrestrial paleognaths and diving loons and grebes using similar analysis make the authors appropriately circumspect of their results.

699 – Presbyornis, Anatalavis and to some extent Conflicto can all be characterized as having a spatulate beak, but they are each very dissimilar from one another and each more alike Stictonetta, Spatula, and Mergus, resp.

768, 811 – Conflicto is reported by Tambussi et al to be Paleocene. No more recent publication is cited to justify citing it as Cretaceous. It is unjustifiable to assume that “Cretaceous” (also Paleocene) Anatalavis had a spatulate bill. Olson, himself, said it would have been impossible to have known that Presbyornis did until its skull was discovered.

814 – gastornithids is an adjective and not capitalized

Figure 2 – The figure would benefit from additional labeling. Given the detailed description of cranial characters that cannot be discerned from the figure, there is an argument to be made to have an artist’s three-dimensional reconstruction of the skull. For a long time, I puzzled over two parallel bars of bone oriented rostrocaudally in the crushed upper bill. One of these must be the right tomium, I concluded that the other must be the left side of the mandible. But if so, then why is there apparently no elevation of the left coronoid as it so clearly is on the right side? There are various seemingly extraneous lines in panel A that appear to be artifacts. If they aren’t, then they should be labeled.

Figure 3 – the furcula of Paakniwatavis (fig. 3 N, O) appears no thicker than that of Presbyornis (fig. 3 M) as claimed

Figure 4 - the labeling F1-F4 is strangely inconsistent with the rest of the figure. More importantly, the caption does not identify whether the images of F1,F2 are of one tarsometatarsus from opposite sides or both right and left. F3,4 are said to be of “the same tarsometatarsus”

Figure 5 – the reconstruction is aesthetic, but somewhat misrepresentative, i.e., unfused lacrimals (not described, if so), sternum tiny, furcula narrow, the angle of dorsal and sternal costae oriented as in flightless ‘ratites’, intramembral proportions of hindlimb elements not quite as reported on lines 235, 362. Missing cervical vertebrae are indicated by dark gray. Other missing parts of the skeleton (e.g., of the sternum, proximal left tibiotarsus, ) should be, too, for consistency although much of that can be gleaned from the line drawing background of the figure.

There are five blank pages following character description #719 in the Appendix. Were there supposed to be illustrations or citations here?

The number of characters scored for this study are impressive, but relatively few can be scored for the new fossil. The resulting phylogeny might therefore be robust for neotaxa (however, the much larger dataset of Livezey and Zusi failed to reproduce what are known relationships even of living birds), but have little to say about the position of this fossil within it. Despite the scoring of ten “relevant traits that detail habitat preference” it is a stretch of credulity to suggest that ancestral state reconstruction could meaningfully predict the status of characters such as rhamphothecal lamellae in what is a marginally fossil, even by Green River standards.

Unfortunately, I could not verify many of the most important character descriptions because the data were not made available to me. The authors state that the data will become available on Morphobank and Dryad following acceptance and supply of a manuscript number. This is inconsistent with the registration of LSIDs in which the paper was reported to be “in press” in PloS One (notably before I received a request to review it).

One of my biggest concerns is whether the name Paakniwatavis is available or preoccupied. The authors used the name, together with a full description of the fossil in a preprint that appears on BioRxiv. Unrelated to the present manuscript, in May 2022 I inquired with the Secretary of the ICZN whether this could be done. I was advised not to do so. In December 2022 I became aware of a preprint of the present manuscript on bioRxiv in which the name was erected and assigned an LSID. There are only two other LSIDs associated with BioRxiv registered on Zoobank. Both are published works. I queried the Administrator of ZooBank whether Paakniwatavis was preoccupied or a nomen nudum. His answer was that it is unclear. This is his response:

“As for bioRxiv being pre-publication by definition, this is something the Commission has discussed and has not yet resolved. Art. 9.9 says that works cannot be published as “preliminary versions of works accessible electronically in advance of publication”. However, this Article is very problematic because it simply says that a work cannot be published until it is published. And technically, fulfilling all the requirements of the Code for a published work means it’s published. If an electronic work fulfills all the criteria for publication (e.g., prior registration in ZooBank, evidence of registration within the work itself, online archive and ISSN indicated in ZooBank, etc.), then it is published in the sense of the Code. The main reason why bioRxiv would fail the criteria for publication would be Art. 8.1.1: “it must be issued for the purpose of providing a public and permanent scientific record”. So, if bioRxiv is not explicitly indicated as issuing PDFs for the purpose of providing a public and permanent scientific record”, then it would fail to serve as a venue for Code-compliant published works. However, there is no guidance in the Code on how one assesses whether a given PDF is issued for the purpose of providing a public and permanent scientific record. Unless bioRxiv explicitly states that this is not it’s purpose, one could argue that it is intended for the purpose of providing a public and permanent scientific record. … In summary, nothing has yet been published in the sense of the ICZN Code within bioRxiv. And there is some debate whether anything *could* be published therein (based on interpretations of Art. 8.1.1, and Art. 9.9). … as there is no clear answer for how to interpret Art 9.9 (the rule that prevents publication as “preliminary versions of works”). Thus, it’s somewhat ambiguous whether names created within a “pre-publication” can be Code-compliant – depending on how one interprets Art. 9.9 and Art. 8.1.1). … This is a really interesting case, because technically, by a very narrow interpretation of the Code (which I doubt most Commissioners would agree with), it fulfills the requirements of the Code, unless the PDF version available on bioRxiv (https://www.biorxiv.org/content/10.1101/2022.11.23.517648v1.full.pdf) fails Art. 8.1.1. I will need to share this example with the Commission to see if there is any way it could be considered published in the sense of the Code as a PDF available through bioRxiv. I will need to get back to you on this.” [He has not, but in a subsequent email wrote:] “I’m still drafting the email to the Commission to examine this case. I also just noticed your email to me from two days ago – VERY sorry I missed that! I would have replied sooner had I not missed it! Indeed, this is a very interesting case.”

6. PLOS authors have the option to publish the peer review history of their article (what does this mean?). If published, this will include your full peer review and any attached files.

Reviewer #1: No

Reviewer #2: No

---

## [Author Response · Author response to Decision Letter 0]

31 Mar 2023

Please see the attached Response to Reviewers document and the Cover Letter.

---

## [Decision Letter · Decision Letter 1]

29 May 2023

PONE-D-22-32093R1A new Paleogene fossil and a new dataset for waterfowl (Aves: Anseriformes) clarify phylogeny, ecological evolution, and avian evolution at the K-Pg Boundary.

PLOS ONE

Dear Dr. Musser,

Thank you for submitting your manuscript to PLOS ONE. After careful consideration, we continue to believe that it has merit but does not fully meet PLOS ONE’s publication criteria as it currently stands. Therefore, we invite you to submit a new revised version of the manuscript that addresses the points raised during the review process.

Both reviewers have accepted to comment on your revised manuscript. They provide very detailed analysis of the manuscript, especially reviewer 2, who put a lot of efforts to provide abundant useful information about morphological and phylogenetic characters, and bibliographic references that were requested in the first round of reviews. They recognize that the manuscript represents a significant investment of work, but it still leaves several unaddressed issues in its current form. Several confusions must also be clarified in your revision before any acceptance.

Reviewer 2 made his comments in a document that is attached. Please be sure to download it.

We look forward to receiving your revised manuscript.

Kind regards,

Thierry Smith, Ph.D.

Academic Editor

PLOS ONE

Reviewers' comments:

Reviewer's Responses to Questions

**Comments to the Author**

1. If the authors have adequately addressed your comments raised in a previous round of review and you feel that this manuscript is now acceptable for publication, you may indicate that here to bypass the “Comments to the Author” section, enter your conflict of interest statement in the “Confidential to Editor” section, and submit your "Accept" recommendation.

Reviewer #1: (No Response)

Reviewer #2: (No Response)

2. Is the manuscript technically sound, and do the data support the conclusions?

Reviewer #1: Yes

Reviewer #2: Partly

3. Has the statistical analysis been performed appropriately and rigorously? 

Reviewer #1: Yes

Reviewer #2: No

4. Have the authors made all data underlying the findings in their manuscript fully available?

Reviewer #1: (No Response)

Reviewer #2: No

5. Is the manuscript presented in an intelligible fashion and written in standard English?

Reviewer #1: Yes

Reviewer #2: Yes

6. Review Comments to the Author

Reviewer #1: You correctly note that I confused your phylogenetic placement in my previous comments for which I apologize. However, this confusion stems from your statement in the abstract (line 56) that says "We recover the new Eocene taxon as a stem anseranatid across all analyses". In line

696 you also note that seven characters "support placement of P. grandei as a stem anseranatid". In your response to the reviewers you also note that the figures of the "resulting trees clearly show that we recover Paakniwatavis grandei as a stem anseranatid across all analyses". I find this very confusing. In the tree, the new taxon is the sister group of the clade (Anseranatidae + Anatidae) - do you use the term Anseranatidae for this clade (which would be very strange and conflict with your use of Anseranatidae in the figures), or is this a lapsus? Based on the phylogenetic trees, the new fossil is a stem group representative of the clade (Anseranatidae + Anatidae), not a stem anseranatid.

My comments on the "duck-like" beak were based on my erroneous assumption that you found the new taxon to be within the clade (Anseranatidae + Anatidae), as a "stem anseranatid" (as repeatedly note in the paper). In this case, the presence of a galliform-like beak would conflict with the duck or goose-like beaks of anseranatids and anatids. Of course, my comments is not valid if the new taxon is outside the clade (Anseranatidae + Anatidae).

Your statement that “Seven unambiguous and three ambiguous optimized synapomorphies of the quadrate, mandible, furcula, scapula, carpometacarpus, tarsometatarsus and pedal phalanges support placement of P. grandei as a stem anseranatid” is not a standard one used in phylogenetic analyses. Apart from the stem anseranatid vs. stem (anseranatid + anatid) issue, your characters can only support an assignment of the taxon to a certain clade. In order to show that it belongs to the stem group of this clade, you would need to list characters that diagnose the crown group of this clade and that are absent in the fossil.

In my first review, I have also commented on screamer-like fossil from North America and Europe, which you omitted from your manuscript. These have meanwhile been described as Anachronornis and Danielsavis: Houde, P., Dickson, M., & Camarena, D. (2023). Basal Anseriformes from the Early Paleogene of North America and Europe. Diversity, 15(2), 233.

In this latter paper, comparisons with the Green River Fossil (Paakniwatavis) were made, and I think you should at least briefly address this study in your paper. On the one hand, the study addresses the fossil you describe, on the other it describes fossils that are possibly related to your new species. Your new taxon also needs to be diagnosed from Anachronornis and Danielsavis.

In your response to the authors you note that Gastornis is a junior synonym of Diatryma. This is certainly not true: Gastornis was described by Hebert 1855 whereas Diatryma is authored by Cope 1876. Gastornis therefore clearly precedes Diatryma by more than two decades.

Reviewer #2: Please refer to attachment for my full review. I paste the first paragraph only here to fulfill the minimum 100 character requirement.

Every novel early Paleogene bird fossil is important in its own right, and this manuscript represents a significant investment of solid work to describe one. That being said, it is unfortunate that the revised manuscript still leaves a great deal to be desired in its current form. The most fundamental among its conceptual flaws are overstated conclusions drawn from ancestral state reconstruction that itself is dependent on a suspect phylogeny. The many origins of these flaws are addressed below.

Continued in attachment.

7. PLOS authors have the option to publish the peer review history of their article (what does this mean?). If published, this will include your full peer review and any attached files.

Reviewer #1: No

Reviewer #2: No

---

## [Author Response · Author response to Decision Letter 1]

7 Sep 2023

Please see uploaded PDF titled "Response to Reviewers."

---

## [Decision Letter · Decision Letter 2]

13 Oct 2023

PONE-D-22-32093R2A new Paleogene fossil and a new dataset for waterfowl (Aves: Anseriformes) clarify phylogeny, ecological evolution, and avian evolution at the K-Pg Boundary.

PLOS ONE

Dear Dr. Musser,

Thank you for submitting your manuscript to PLOS ONE. After careful consideration, we feel that it has merit but does not fully meet PLOS ONE’s publication criteria as it currently stands. Therefore, we invite you to submit a revised version of the manuscript that addresses the points raised during the review process.

The reviewers have accepted to comment a third time on your revised manuscript. They both consider that this is a very important fossil that needs to be published. However, they both still think the manuscript has many issues. They clearly state where are the issues and offer detailed answers to help improving the manuscript.

May I please you to follow the comments of the reviewers and take actions to make the requested changes. Additional work is required to improve the manuscript.

I really hope you can address the comments, improve substantially the manuscript, and then I would be glad to accept the manuscript.

We look forward to receiving your revised manuscript.

Kind regards,

Thierry Smith, Ph.D.

Academic Editor

PLOS ONE

Reviewers' comments:

Reviewer's Responses to Questions

**Comments to the Author**

1. If the authors have adequately addressed your comments raised in a previous round of review and you feel that this manuscript is now acceptable for publication, you may indicate that here to bypass the “Comments to the Author” section, enter your conflict of interest statement in the “Confidential to Editor” section, and submit your "Accept" recommendation.

Reviewer #1: (No Response)

Reviewer #2: (No Response)

2. Is the manuscript technically sound, and do the data support the conclusions?

Reviewer #1: Partly

Reviewer #2: Partly

3. Has the statistical analysis been performed appropriately and rigorously? 

Reviewer #1: N/A

Reviewer #2: I Don't Know

4. Have the authors made all data underlying the findings in their manuscript fully available?

Reviewer #1: Yes

Reviewer #2: No

5. Is the manuscript presented in an intelligible fashion and written in standard English?

Reviewer #1: Yes

Reviewer #2: Yes

6. Review Comments to the Author

Reviewer #1: (1) I strongly suggest to restructure the Differential Diagnosis and to better delimit the new taxon from Danielsavis and Anachronornis.

You start by saying "The rostrum shape exhibited by this taxon is unlike most other previously recovered

Paleogene Anseriformes". However, then you rightly proceed by saying that "When compared to extinct taxa, the narrow bill of Paakniwatavis is most like those of Anachronornis anhimops and Danielsavis nazensis". Most of the comments pertaining to Danielsavis and Anachronornis in your Diff Diagnosis refer to the fact that comparisons of certain elements are not possible, and what would be needed is the inclusion of two paragraphs, in which the new taxon is clearly differentiated from Danielsavis and Anachronornis (note that the wider coracoid shaft may be the result of flattening of the GRF specimen). Actually, a differentiation from Danielsavis and Anachronornis is key to your manuscript, since these two taxa certainly are most similar to your new fossil.

In a differential diagnosis you should actually list similar taxa and differentiate the fossil from the by saying ABC differs from XYZ in .... A diagnosis, by contrast list characters that are considered distinctive (ideally autapomorphic) for a new taxon. Your diagnosis is confusing, because you mix both approaches and make comparisons by skeletal elements, not by taxa. Instead of writing feature XYZ is like in ABC but different from DEF, it would be much clearer to structure the Differential Diagnosis by differentiating Paakniwatavis from each taxon separately (mainly Anachronornis, Danielsavis, and Nettapterornis)

You use the bill width of the new taxon in the differential diagnosis and say that "Paakniwatavis grandei exhibits a mediolaterally narrow bill that is narrower than the width of the skull at the parocciptal processes". However, this statement is not quite clear and needs to be specified further. On the one hand, the skull is exposed in lateral view, so I do not quite understand how you can infer bill width. On the other hand, from what is preserved, the beak appears to have been tapering towards its tip, so that that its width was not constant (the base certainly was wider than the tip).

As a side note: I do not think that the Anseranas-like skull of your reconstruction matches the preserved outline of the skull of the fossil, in which the beak appears to be more anhimid-like. In line 102, you note that "the new fossil presents a narrow bill that is most similar to the Anhimidae" - why, then, has the beak be reconstructed as Anseranas-like (dorsoventrally narrow, hooked tip etc.)?

It is also a misleading that your reconstruction shows a crest or feather tuft on top of the skull - I see no evidence fort this in the fossil, which does not exhibit any soft tissue preservation. This crest/tuft serves to make the reconstruction more Anseranas-like than what is supported by the actual fossil, which likewise shows no evidence for the length of the tail feathers.

(2) Please also note that the Danielsavis material has meanwhile been redescribed and additional features (such as the palate and the pedal phalanges) having been described. Danielsavis was found to be very different from Anachronornis and may actually be a stem group galliform not an anseriform; it is now assigned to a new taxon Danielsavidae.

Mayr, G., Carrió, V., & Kitchener, A.C. 2023: On the “screamer-like” birds from the British London Clay: An archaic anseriform-galliform mosaic and a non-galloanserine “barb-necked” species of Perplexicervix. Palaeontologia Electronica 26(2):a3328; doi: 10.26879/1301.

I agree with the comments of the other reviewer that it would be appropriate to note that Grande (2013) compared the fossil with the Anhimidae. Even if this identification may go back to an initial identification by one of the authors of the present manuscript, it is a published statement upon which future readers may stumble.

- line 302: I agree with reviewer 2 that a distally "hooked" hypotarsus is not visible on the published figures

- line 596: sp. nov. (instead of "sp. Nov.")

- Line 827: "Anachronornis anhimops and Danielsavis nazensis were largely placed outside of

Paleognathae" - I do not quite understand why this statement is here, since none ever suggested that these birds are palaeognathous. Note also that it is "Palaeognathae".

Reviewer #2: Review of PONE-D-22-32093R2 A new Paleogene fossil and a new dataset for waterfowl (Aves: Anseriformes) clarify phylogeny, ecological evolution, and avian evolution at the K-Pg Boundary.

59-60: citation 11 recovers Pelagornithidae as neoavian, not anseriform, as does more recent analysis by Benito et al, cited elsewhere in the manuscript but not cited here. The disagreement among analyses is discussed much later (678-692) but is misrepresented here that there is no disagreement. It should be.

62: Anhimidae are aquatic foragers, not terrestrial except maybe in captivity

73: the term vestigial explicitly refers to an organ that is derived from a functional one but that is no longer functional. This is not the same as the term rudimentary, which does not imply character state polarity. The use of term vestigial is appropriate in the context used here (because this had been suggested long ago) but inappropriate elsewhere in the manuscript.

73-74: anhimids are not palmate, semipalmate at best. This is in fact conceded elsewhere in the manuscript.

81: either misplaced period or capitalize “and”

103, 227: Mayr et al (2023; https://doi.org/10.26879/1301; not cited) vitiated the suggested anseriform relationship of Perplexicervix

206 Diagnosis, 241: Unfortunately, Anachronornis is insufficiently preserved (lacks complete tibiotarsus and tarsometatarsus) to distinguish from Paakniwatavis based on these diagnostic criteria, but it is obviously a different animal based on other characters. Perhaps one or more some should be mentioned in the Diagnosis. I am not fully convinced that the crushed furcula is thick as described based on figure 3N and 3O. It could be, but it doesn’t look different than that of Presbyornis in fig. 3M.

224: add space “and3”

230: Explain what is meant by “the facial margin” of the bill or use accepted terminology. It isn’t clear to me.

233: relative lengths of external nares and rostrum of Anachronornis and Danielsavis are published (Houde et al 2023 figs. 1D and 7; Mayr et al 2023 fig 1B) so this statement is incorrect

249: Houde et al specifically state the humerus of Anachronornis is apneumatic (Houde et al p 19 “Humerus: Nonpneumatic.”) so this statement is incorrect

254: misplaced “I”

258: do the authors mean metacarpals instead of digits? There are no synostoses in the digits (i.e., phalanges). If the carpometacarpus is intended here, then they should also indicate that this is in reference to the distal synostosis.

281-282: “pedal phalanx IV: digit IV is longer than IV: III” makes no obvious sense because the fourth phalanx in digit III is a claw and in digit IV it is not. Perhaps the words digit and phalanx are transposed.

283-288: the description goes from carpometacarpus to humerus back to carpometacarpus. This disorganization was commented on in a previous review.

318-319: no aspect of the bill even remotely resembles Anseranas except that the caudal portion (only) of the tomium is straight. This was commented on in a previous review. The authors are unwilling to concede the obvious.

330-331: I have been provided with no image to verify that the postorbital process was long as claimed. It would be surprising if it was long, since it is not in either anhimids or Presbyornis. This was commented on in a previous review.

379: It is ambiguous whether the authors are stating that the vertebrae of Presbyornis, Nettapterornis and Telmabates are or are not heterocoelous.

381: As above. It is less ambiguous to say “The notarium is present in all Anseriformes…”

431: forms

454: nix “appears”

455: which of the two cnemial crests, cranial or lateral?

517: segmented files are not available on morphobank as claimed. There is one obj file of the quadrate and one stl file of a tarsometatarsus, nothing more. None of the descriptions and character scoring based on segmented CT scans other than these two are verifiable by readers. They need to be.

521-526: There are only 41 taxa included in the nex files in morphobank. In other words, the matrix provided does not include Anachronornis, Danielsavis and Perplexicervix as claimed. Just as importantly, the matrix includes no unambiguous Neoaves. The importance of this is that there are no data included to challenge the recovery of Pelagornithidae within Galloanseres, which is a viable and the most current hypothesis of their relationships.

571: In a previous review, objection was raised about the inclusion of the PCA and LDA analyses on the grounds that they recovered results known to be incorrect even for neotaxa. The authors dismissed this concern. They stated that they agreed and therefore moved these analyses to the Supplementary Data. Maybe so fig S4, but in fact they retained these in the Methods and in the Results section 778-794 and the results are alluded to in the Discussion in the form of interpretations. It should be deleted altogether.

575: Nettapterornis is labeled Anatalavis in the PCA figure S4.

582-584: the authors state in their response to reviewers that Paakniwatavis is recovered as sister to Anseranatidae plus Anatidae, but here it clearly states “Paakniwatavis … is recovered across all analyses as the sister taxon of Anseranatidae.” What’s more, if sister to Anseranatidae plus Anatidae, then in fig. 5 Anatidae includes Vegavis, Presbyornis, Conflicto, Telmabates, and Nettapterornis. That is going to raise some eyebrows.

536: delete “.;”

537: What “iteration”? Do the authors simply mean they adopted characters from other datasets? If so, then just say so.

552 and 556 are redundant.

585: both?

597: see comments about fig. 5

603-604: is somewhat redundant with 673-674 with regard to Dendrocygna and Thalassornis

611: fhigh

643-645: Rewrite sentence. Did Sun et al use mitochondrial genomes or two genes? Does “which contains almost all extant anseriform taxa used in our analysis” mean that there was sequence data for all the extant species used in Musser and Clarke’s analysis? This is confusing.

646: delete “an”

652, 673, 674: Thalassornis is misspelled

678-683: run-on sentence would be better broken into two, first reporting the results and the second the explanation for the low support.

688-709: the Results section should report the results of this study, but it includes considerable text reviewing the results of other peoples’ analyses. Those ought to go into the Discussion.

696 and elsewhere: anatid should be capitalized only when expressed as the formal name Anatidae

730: to say the crest is prominent obviates the need to say it is present

744-777: I like the two paragraphs on ancestral state reconstruction. It is candid and appropriately circumspect. Unfortunately, the same cannot be said for how these traits are mapped onto the phylogeny in figure 5 and how they might be misinterpreted by readers who do not pay sufficient attention to these two paragraphs. Please see my comments on fig 5 below.

752-754: both some jacanids and anhimids have third digits that are longer than the tarsometatarsus (specimens directly available to me, but consult Ridgeway for individual variation). Both are renowned for walking on floating vegetation, not swimming (although screamer chicks notably swim), for which the long toes are specialized. Ref 106 is irrelevant to the authors’ point since it concerns claw curvature, not digital length.

778-794: this paragraph really needs to be deleted, as does figure S3, as articulated in previous review. Even the authors concede “the results are largely inconsistent with other features or the known behaviors [of neotaxa] (i.e., Chauna)”. It is nonsense, so don’t include it.

795: At the risk of sounding gratuitously critical, the writing could be much improved. Much of the discussion is rambling, verbose, rhetorical, speculative, redundant, tangential or irrelevant, and conversational. It reads like a first draft. It would benefit from being shortened with an eye to the elements of style. There are excellent books on the subject that should be read by all whose careers depend on effective communication.

799-800: The relationships of Pelagornithidae and Gastornis are not tested in this study because their purported neoavian sisters were not included in the dataset. It is meaningless to report where they are recovered among taxa that do not represent the diversity of current alternate hypotheses of their relatives. There are three ways to deal with this. First, add representative Neoaves to the dataset and rerun the analyses. Most of these data are already scored in the datasets of the studies from which the current matrix was derived. This would be a valuable exercise even if some data were missing. Moreover, they can be part of a backbone constraint tree. Second, Pelagornithidae and Gastornis could be deleted from this dataset and report. Third and by far the least desirable, the authors could include a caveat that states that Pelagornithidae and Gastornis are recovered as … but their proposed neoavian sisters (include refs) were not included in this study. I realize the third option is the easiest for the authors because it requires virtually no effort, but it is the poorest remedy if a true remedy at all.

826-827: the phrase “especially combined with a reduced form of rhamphothecal lamellae present in extant Anhimidae [22]” should deleted because it implies character state polarity of rhamphothecal grooves in Anseriformes that is not supported by the current analysis. In fact, that the grooves are “reduced” is contradicted by the ancestral state reconstruction. “Incipient” might be a more appropriate descriptor than reduced, but that too implies character state polarity that truly is beyond the scope of this study.

838-840: No, Conflicto, Anachronornis, and Danielsavis all have narrow bills and only Presbyornis and Nettapterornis are known to have wide ones. Brief reference to Anachronornis and Danielsavis is appended to the end of this paragraph (857-859) in what seems to be a minimal effort for resubmission without actual revision that brings this manuscript up to date. The bill of Conflicto is narrow and utterly unlike that of Nettapterornis. The bill of pelagornithids is so manifestly different from that of Anseriformes (instead, like those of neoavian Pelecaniformes and Procellariiformes, e.g., in the presence of rhamphothecal segmentation as well as dimensions) that pelagornithids don’t warrant mention here.

860-888: this entire paragraph is speculative story-telling and should be deleted. The argument as stated in 863-865 is flawed because Anhimidae is not nested within Anseranatidae-Anatidae (it is sister to them instead; fig 5) and because screamers are aquatic foragers, not terrestrial. The presence of the diminutive ridges in the rhamphotheca of screamers is not evidence of developed rhamphothecal grooves as an ancestral state in crown Anseriformes and that they are vestigial in screamers. Lines 875-878 are undoubtedly correct, but without requiring a character state reversal in screamers. I don’t think 865-866 is correct. As stated in previous review, ridges exist in the rhamphotheca of at least some Psittaciformes (lacking pedal webbing) for what it is worth. The excessive digression on non-avian dinosaurs and turtles is tangential and a distraction. The fact is that rhamphothecal grooves have been specialized for a variety of diverse foraging behaviors in Anseriformes and others, including filter-feeding in Anatini (and flamingos), piscivory in Mergini, and grazing in Anserini. Last, 887: it is not news that “feeding ecology has acted as the primary selective force in waterfowl beak shape diversification [19].” Feeding ecology is the primary selective force in beak shape of all birds, without exception. Thermoregulation, display, and phylogenetic inertia play secondary roles in some.

916-918: This sentence adds nothing

920: Dromornithids were not included in this study, so why are they included in “this clade”?

919-927: this paragraph should be reorganized with a topic sentence that the thoracic vertebrae of Paakniwatavis are non-heterocoelous, but distorted. Then, go on to explain why in the context of others.

938: K-Pg, Tertiary is obsolete and the term Paleogene is used elsewhere in this manuscript

953: pseudodontorns

975-979: run-on sentence about calibration points morphs into a string of allusions to preservation, phylogeny, ecology, behavior and biogeography.

Figure 5 needs to be deleted and redone from scratch. Taxa are omitted from the phylogeny. The artist’s reconstruction of Paakniwatavis is patently misleading with regard to the skull, sternum, costae, as detailed in previous review. The ancestral state reconstruction does not discriminate between faint ridges on the palate of screamers (and other birds unmentioned) and the pronounced and highly specialized rhamphothecal lamellae of extant anatids, nor between the semipalmate digits of Anseranas and the fully palmate digits of anatids. It also invokes the evolution of said ‘webbing’ after the divergence of anhimids and Paakniwatavis, contrary to the text.

The provided dryad link returns the message “the page you are looking for does not exist.” I had to request an alternate working link by an Editor of PLoS One. Therein exist raw unsegmented CT slices, and far fewer than the thousands one would expect. These files might be useful for the rare individual who might want to expend the significant time and effort to personally segment the images from scratch, and as such they appropriately need to be deposited and made available. However, they will be of no interest to nearly all readers who will want to verify character descriptions and scoring of segmented structures in the form of stl files. The stl files are not available and they must be.

Fig S4: Anatalavis is labeled. Nettapterornis?

All one need do is scan the tracked changes in manuscript R1 to see that the only substantive changes in R2 are the addition of paragraphs about Anachronornis, Danielsavis, and Perplexicervix, as though this was the only thing that was requested by reviewers. The authors dispute nearly every comment by reviewer 2, rather than conceding that any might be well founded. Furthermore, the updated matrix including Anachronornis, Danielsavis, and Perplexicervix that is alleged (523-525) to be deposited in morphobank is not in fact present in the morphobank folder that is linked to this submission.

I can understand the frustration expressed by the authors in the response to reviewers when they say that they believe they uploaded all the relevant supplementary materials but the reviewer says otherwise. But for whatever reason, the referenced files (i.e., updated matrix and stl files of the specimen) do not exist in morphobank or dryad or the supplementary document. The files that do exist were certainly created by the authors, so it seems possible that if these were updated then the updates were not saved or the proper links to them were not passed along to the reviewers.

Paakniwatavis is a very important fossil that needs to be published. “A tenet of scientific inquiry is reproducibility”( Steenwyk et al 2023 https://doi.org/10.1038/s41576-023-00620-x). In its present form, this manuscript does not permit reproducibility. I cannot independently verify the results reported in this manuscript without necessary image files and the complete data matrix.

7. PLOS authors have the option to publish the peer review history of their article (what does this mean?). If published, this will include your full peer review and any attached files.

Reviewer #1: No

Reviewer #2: No

---

## [Decision Letter · Decision Letter 3]

10 Mar 2024

PONE-D-22-32093R3A new Paleogene fossil and a new dataset for waterfowl (Aves: Anseriformes) clarify phylogeny, ecological evolution, and avian evolution at the K-Pg Boundary.PLOS ONE

Dear Dr. Musser,

Please find here the new comments of the reviewers on the basis of your last revision. Both reviewers highlight the significant improvements and key issues you addressed in the present revision. They both provide here few last points that still have to be addressed regarding editorial corrections and references. Please pay attention that the reviewer 2 provides his comments in a separated file, which is in attachment.

Regarding some scientific issues, reviewer 2 put again a lot of efforts to help improving the manuscript and come back to points that have not been addressed in previous revisions. For most points he proposes solutions. Among these points are issues that I consider as particularly important for acceptance, especially regarding PLOS ONE's policy:

- Issues related to character scoring in the phylogeny.

- Characters that are discussed but not verifiable on the present figures.

We invite you to submit a revised version of the manuscript that addresses the points raised during the review process.

I wish to encourage you to pursue your efforts of improving this manuscript, hoping that soon I can definitely accept it.

We look forward to receiving your revised manuscript.

Kind regards,

Thierry Smith, Ph.D.

Academic Editor

PLOS ONE

Journal Requirements:

Reviewers' comments:

Reviewer's Responses to Questions

**Comments to the Author**

1. If the authors have adequately addressed your comments raised in a previous round of review and you feel that this manuscript is now acceptable for publication, you may indicate that here to bypass the “Comments to the Author” section, enter your conflict of interest statement in the “Confidential to Editor” section, and submit your "Accept" recommendation.

Reviewer #1: (No Response)

Reviewer #2: (No Response)

2. Is the manuscript technically sound, and do the data support the conclusions?

Reviewer #1: Yes

Reviewer #2: Partly

3. Has the statistical analysis been performed appropriately and rigorously? 

Reviewer #1: Yes

Reviewer #2: No

4. Have the authors made all data underlying the findings in their manuscript fully available?

Reviewer #1: Yes

Reviewer #2: No

5. Is the manuscript presented in an intelligible fashion and written in standard English?

Reviewer #1: Yes

Reviewer #2: Yes

6. Review Comments to the Author

Reviewer #1: I am satisfied to see that the authors finally addressed some of the key issues raised in previous review and only have a few final points that should easily be addressed:

- You have noted that you included Mayr et al. 2023 in your revised description, but neither has the paper been cited nor do I find any reference to it. This is unfortunate, because Mayr et al. (2023) figured and described various elements that were not figured by Houde et al. (2023) and that could be used to differentiate your new taxon (e.g., there are differences in the lengths of the pedal phalanges).

In the comparisons in line 362 ff. you seem to refer to the coracoids figured by Mayr et al. (2023) and this should be mentioned, because in none of the cited publication the coracoid of the taxon has been figured.

- In the systematic heading: please provide the author/publication year for Anseriformes and format the heading according to the other headings

- Throughout the manuscript: it is "impressio musculi"/"impressio m." (not "impression musculi")

- lines 239 and 361 (and perhaps elsewhere): write Paakniwatavis in italics

- line 364: Perplexicervix (not Perplexicervis)

Reviewer #2: I congratulate the authors for making numerous significant improvements in the current revision, particularly with regard to Figure 5. I hope that it is evident from the effort I have invested to make even minor editorial corrections, which themselves could not possibly influence a final decision, that my criticisms have always been intended to be constructive. Paakniwatavis is an important fossil, deserving of the best possible description. I hope the authors and Editor(s) take this to heart as they read my comments below. I apologize in advance for the length and some redundancy. This is a long and complicated paper that warrants it.

Please refer to attachment for remainder of review.

7. PLOS authors have the option to publish the peer review history of their article (what does this mean?). If published, this will include your full peer review and any attached files.

Reviewer #1: No

Reviewer #2: No

---

## [Author Response · Author response to Decision Letter 3]

24 Apr 2024

Please see uploaded documents for response to reviewers and notes to editor.

---

## [Editor Report · Decision Letter 4]

8 May 2024

A new Paleogene fossil and a new dataset for waterfowl (Aves: Anseriformes) clarify phylogeny, ecological evolution, and avian evolution at the K-Pg Boundary.

PONE-D-22-32093R4

Dear Dr. Musser,

We’re pleased to inform you that your manuscript has been judged scientifically suitable for publication and will be formally accepted for publication once it meets all outstanding technical requirements.

Kind regards,

Thierry Smith, Ph.D.

Academic Editor

PLOS ONE

---

## [Editor Report · Acceptance letter]

20 May 2024

PONE-D-22-32093R4 

PLOS ONE

Dear Dr. Musser, 

I'm pleased to inform you that your manuscript has been deemed suitable for publication in PLOS ONE. Congratulations! Your manuscript is now being handed over to our production team.

Kind regards, 

on behalf of

Dr. Thierry Smith 

Academic Editor

PLOS ONE